



# CLIM4OMICS: a geospatially comprehensive climate and multi-OMICS database for Maize phenotype predictability in the U.S. and Canada

Parisa Sarzaeim[1], Francisco Muñoz-Arriola[1,2], Diego Jarquin[3], Hasnat Aslam[4], Natalia De Leon Gatti[5]

[1]Department of Biological Systems Engineering, University of Nebraska-Lincoln, Lincoln, NE, 68583-0726 USA, Email: parisa.sarzaeim@huskers.unl.edu
[2]School of Natural Resources, University of Nebraska-Lincoln, Lincoln, NE, 68583-0996 USA, Email: fmunoz@unl.edu
[3]Agronomy Department, University of Florida, Gainesville, FL, 32611 USA, Email: jhernandezjarqui@ufl.edu
[4]School of Natural Resources, University of Nebraska-Lincoln, Lincoln, NE, 68583-0996 USA, Email: haslam2@huskers.unl.edu
[5]Department of Agronomy, University of Wisconsin-Madison, Madison, WI 53706 USA, Email: ndeleongatti@wisc.edu

*Correspondence to*: Francisco Munoz-Arriola

**Abstract.** The performance of numerical, statistical, and data-driven diagnostic and predictive crop production modeling heavily relies on data quality for input and calibration/validation processes. This study presents a comprehensive database and the analytics used to consolidate it as a homogeneous, consistent, and multi-dimensional genotype, phenotypic, and environmental database for maize phenotype modeling, diagnostics, and prediction. The data used is obtained from the Genomes to Fields (G2F) initiative, which provides multi-year genomic (G), environmental (E), and phenotypic (P) datasets that can be used to train and test crop growth models to understand the genotype by environment (GxE) interaction phenomenon. A particular advantage of the G2F database is its diverse set of maize genotype DNA sequences (G2F-G), phenotypic measurements (G2F-P), station-based environmental time series (mainly, climatic data) observations collected during the maize growing season (G2F-E), and metadata for each field trials (G2F-M) across the U.S. and the province of Ontario in Canada. The construction of this comprehensive climate and genomic database incorporates the analytics for data quality control (QC) and consistency control (CC) to consolidate the digital representation of geospatially distributed environmental and genomic data required for phenotype predictive analytics and modeling the GxE interaction. The two-phase QC-CC pre-processing algorithm also includes a module to estimate environmental uncertainties. Generally, this data pipeline collects raw files, checks their formats, corrects data structures, and identifies and cures/imputes missing data. This pipeline uses machine learning techniques to fulfill the environmental time series gaps and quantifies the uncertainty introduced by using other data sources for gaps imputation in G2F-E, discards the missing values in G2F-P, and removes rare variants in G2F-G. Finally, an integrated and enhanced multi-dimensional database is generated. The analytics for improving the G2F database and the improved database called "CLIM4OMICS" follows the FAIR principles, and all the digital resources are available at http://doi.org/10.5281/zenodo.7490246 (Sarzaeim, et al., 2023).



## 1. Introduction

The evolving nature of the Earth System models, Artificial Intelligence, and data availability requires a more comprehensive suite of analytics for quality and consistency controls (Livneh et al., 2015; Reyer et al., 2020; Quiñones et al., 2021) that foster
the democratization of data collection, management, transformation, and adoption FAIR principles. In this changing digital environment, data quality and uncertainty assessment on the train and test datasets become critical to improve models' performance and ability to predict systems of natural and human origin (Furche et al., 2016; Jiang et al., 2017; Sarzaeim et al., 2022a). We introduce the analytics for quality, and consistency controls useful for the development and consolidation of an enhanced, high-quality, large-scale, and multi-dimensional database for maize phenotype predictability using genomics and
phenomics (OMICs) data and meteorological and climatological observations distributed across maize production areas in the U.S. and a province in Canada.

The creation of multi-dimensional databases consistently grapples with integrating the multiple sources and spatiotemporal attributions of data, including variety, velocity, volume, and other seven characteristics known as "Vs" of Big Data (Firican, 2017; Janev, 2020). Exploration, discovery, planning, and management of biological systems under volatile and unevenly
distributed climate conditions favor the collection, transfer, transformation, and construction of multi-dimensional databases with disparate structures and uncertainties (Gonzalez-Rouco et al., 2001; Hubbard et al., 2005; Brönnimann et al., 2006; Sertel et al., 2010; Chiu et al., 2009; Sarzaeim et al., 2022a). The use of accessible analytics for quality and consistency controls for a growing availability of OMICs including climate data becomes critical for creating and making valuable databases, democratizing data construction, access, improvement, and using data for discovery and innovation (Overpeck et al., 2011;
Shekhar et al., 2017; OKN-NSF, 2022).

Generally, quality control (QC) frameworks are characterized by the identification of technical errors in data collection (Livneh et al., 2015) and the diagnostics and removal of data outliers (Gonzalez-Rouco et al., 2001, Alkhalifah et al., 2018). Habib et al. (2010) described QC as a process designed to check the correctness and completeness of models' input data. QC is traditionally oriented to detect and discard erroneous samples, decreasing uncertainties in model outputs. For example, Chiu
et al. (2009) employed QC based on geospatial interpolation to identify missing data and eliminate erroneous values in a dataset of geospatially and heterogeneously distributed meteorological stations. While the heterogeneity of spatially distributed data is critical, temporal gaps are an integral part of a robust database for predictive phenotype analytics and models. Lin and Habib (2021) proposed a framework for QC of multi-temporal data for phenotyping from LiDAR, developing external and internal controls to increase accuracy in automated phenotyping. In another study, Wart et al. (2013) applied a QC algorithm to detect
the incorrect temperature, precipitation, relative humidity, and solar radiation values in time series released by NOAA in parts of the U.S. Midwest and replaced the missing values using interpolation techniques. Similar approaches have been developed and operationalized for hydroclimate data (Maurer et al., 2002; Livneh et al., 2013; 2015). The application of QC analytics for high-dimensional databases has been tested in crop models such as the HybridMaize (Wart et al., 2013) and statistical models such as the GxE approach (Sarzaeim et al., 2022a) to predict maize yields. The latter found that improvements in yield



predictability are directly related to data improvements. However, it remains to be seen whether additional improvements in the inputs and the model or the database enhancement based on certain variables can improve the predictability of phenotypes and, eventually, identify the underlying processes that drive it.

On the other hand, the uncertainty in monitoring and sampling and the inconsistency among the collected data structures and formats are other limitations of predictive analytics and models. Zeng et al. (2015) defined consistency control (CC) as an

intercomparison among independent datasets of the same product, leading to possible synergies to enhance the product. The CC contributes to consolidating multi-dimensional climate and OMICs databases with different formats for phenotype simulations. While climate could be considered another component of OMICS, we intentionally listed through the text as climate to differentiate it with the generic term enviromics (Xu et al. 2022). The designed CC checks the intersection among the quality-controlled OMICS and climatic datasets, discard the discontinued data segments containing corresponding missing

values, and synthesizes the remaining consistent datasets ready for crop growth simulation and prediction applications. Several studies underscore QC and CC's critical and complementary roles in improving model prediction accuracy (Feng et al., 2004; Matthews et al., 2013). For example, Hartkamp et al. (1999) showed how the accuracy of agronomic models' output is affected by the input data quality, emphasizing that data QC is a prerequisite for model applications and that the data CC is complementary for successful model operations. The solutions for the incompatibility of input data and their effects on data

availability improvement have been presented in their study to show the critical role of CC and QC practices. Other efforts by Amaranto et al. (2019;2020) illustrate the need for QC and CC data to improve the predictability of variables connected by human or natural origin processes, such as crop evaporative demands and natural and engineered water supplies.

Uncertainty analysis is critical for developing and implementing models and analyzing observations and simulations. Surendran Nair et al. (2012) and Merchant et al. (2017) shed some light on the sources of uncertainty in models' inputs,

structure and parameters, and calibration/validation. Munoz-Arriola et al. (2009), Pogson (2011), Asseng et al. (2013), and Correa-Jaimes et al. (2022) explain that simplifying the models or using variables that represent key complex processes can contribute to explaining the sensitivities in model performance to uncertainties in input data and multiple environmental processes. The integration of multiple variables also represents a challenge for estimating and explaining uncertainties that emerged from, for example, compounded temperature and precipitation, and are affected by sampling density and

interpretation of spatially distributed data (Rehana et al., 2022; Liu et al., 2022). Furthermore, uncertainties associated with climate and crop model performance require data that allow the analyses of error propagation from the inputs to the outputs (Asseng et al., 2013; Amaranto et al., 2020; Sarzaeim et al., submitted). The diagnostic analyses of observed data and the sensitivity of model performance to the uncertainties in the inputs are related to the quality and consistency controls in high dimensional datasets. These relationships also evidence the necessity of expanding input data and quantifying uncertainties to

improve models and model performance for geospatially suitable and reliable applications (Robertson et al., 2014).

In crop phenotype predictability, large-scale and geospatially distributed experiments integrate crop genetics and climate data to map regions suitable to grow and manage resources adaptively to climate and land-use change (Munoz-Arriola et al., 2009; Tang et al., 2012; Rosenzweig et al., 2013; Jarquin et al., 2014; Ruan et al., 2015; Jarquin et al., 2021; Sarzaeim et al., 2022a).



The Genomes to Fields (G2F) initiative is a large-scale effort designed and operated to improve the predictability of maize phenotypes across the U.S. The G2F initiative has released a well-documented, large-scale, and sharable database for maize breeding, capturing the phenotypes in response to genetic improvement and environmental changes (Alkhalifah et al., 2018). The engineers, researchers, and economists interested in understanding the maize genetic functionality across environments can benefit from the G2F database for phenotypic simulation using statistical models including the genotype by environment (GxE) interaction (Lawrence-Dill et al., 2019). The initial implementation of QC in the G2F database aims to remove the outliers (Alkhalifah et al., 2018). However, large-scale enterprises are more likely to expand errors and inconsistencies like missing samples, uneven records, and emerging locations. Additionally, inconsistencies between the collected data structures and format have been maintained rather than the editing for consistency (Alkhalifah et al., 2018). These limitations reduce the advantages of using the G2F database for implementing the GxE models. Consequently, improving the datasets through gap fulfillment and providing a consistent data structure and format is necessary to implement predictive analytics and models adequately. Hence, we use the G2F data to test a quality and consistency control (QC-CC) framework for database improvement and uncertainty quantification in the input data for the predictability of maize yields in the U.S. and Ontario in Canada. The G2F database offers a geospatial and multi-dimensional suite of variables useful to predict maize traits using models including the GxE interaction. It can improve parameterizations of the Earth System and crop models (Rosenzweig et al., 2013; Ruane et al., 2015). The required four-dimensional database for training and testing the GxE models and the output visualization consists of (1) sequences of maize genomic molecular markers for multiple inbred genotypes (G2F-G); (2) observed phenotypic variables (G2F-P); (3) time series of spatially distributed environmental variables for each experimental trial (G2F-E); and (4) metadata for further analytics and geospatial visualization purposes (G2F-M). Figure 1 illustrates a conceptual framework of the quality and consistency control algorithms of the G2F data to build homogeneous, consistent, and multi-dimensional OMICs and environmental time series for maize phenotypes modeling and prediction.

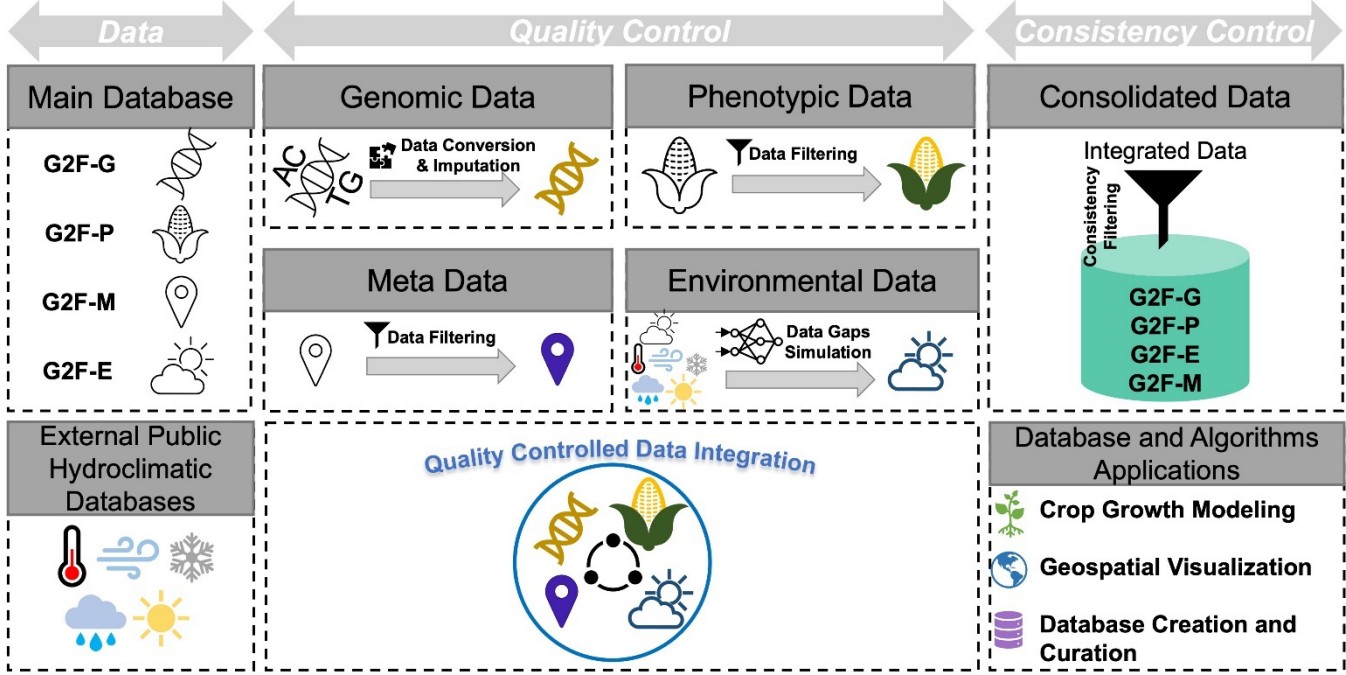

**Figure 1. A conceptual framework of quality and consistency control algorithms for the multidimensional Genomes to Fields (G2F) OMICs and hydroclimatic database. "G2F-G" denotes G2F genomic data, "G2F-P" denotes G2F phenotypic data, "G2F-M" denotes G2F metadata, and "G2F-E" denotes G2F environmental data.**

Open and valid data sources are the foundation for open-source science (Wilkinson et al., 2016; Peng et al., 2022), built upon findability, accessibility, interoperability, and reusability principles, called FAIR data principles (Wilkinson et al., 2016). When these databases follow the FAIR principles, researchers and communities are triggered by the discovery, innovation, and democratization of digital resources (Livneh et al., 2015, Wilkinson et al., 2016, Amaranto et al., 2018, Quiñones et al., 2021, Peng et al., 2022). Nonetheless, the access still exists and limits the user's innovation for more expedited improvements

in data and algorithms for collection-to-curation pipelines. This study consolidates a homogeneous, enhanced, and high-dimensional database following the FAIR data principles for applications in maize breeding and phenotypic modeling and prediction within statistical, data-driven, or biophysical modelling frameworks.

The objectives of this study are to (1) design and develop QC-CC framework to construct an enhanced multi-dimensional database for GxE modeling and geospatial analyses of maize phenotypes predictability; (2) quantify the environmental input

data uncertainties used for maize yield predictions, and (3) provide access to the database and the QC-CC framework pipeline. The study contains six additional sections. Section 2 provides a comprehensive description of the original G2F database, containing a review of each dataset and the associated limitations of the G2F data and metadata. Section 3 contains the foundation and algorithm explanation for the QC module for each dataset (subsection 3.1); the CC algorithm and the compatible multi-dimensional datasets from the quality-controlled data (subsection 3.2); and the quantification of uncertainty



based on the environmental time series errors (subsection 3.3). The results and discussion of the study are presented in Sect.
4. Finally, the data availability statement and concluding remarks are summarized in Sects. 5 and 6, respectively.

## 2. G2F database dimensions

The goal of the G2F initiative is to collect the key datasets to understand roles played by the genotype, environmental
conditions, and agricultural management practices in crops traits (Lawrence-Dill et al., 2019). Since 2014, the G2F initiative
has designed several maize field experiments across the U.S. and Ontario in Canada to integrate a large-scale and multi-
dimensional database required for maize traits prediction. This database provides opportunities for further research and
development in data analytics and different types of modeling approaches for maize phenotype prediction by incorporating
genotype by environment interactions. The G2F platform is updated annually to publish the genomic, phenotypic,
environmental, and metadata collected from the maize field trials. The genomic data is published in one file containing the
molecular markers of all maize inbred lines tested and/or used as parents of the hybrids observed in the G2F sites in the
experimental years. While the phenotypes, environments, and metadata are published in separate annual years. Two released
versions for each phenotypic and environmental data for a given year: (1) raw and (2) clean data files. The raw file is the first
integrative version of the data collected by the G2F collaborators in each experimental site. After implementing initial checks
on the format, data structure, and wrong entries calibration, the clean file is the controlled version of the raw file. This study
uses the clean version files, yet there are still several missing values, typos, and data structure inconsistencies among the clean
version files from different years, which constrain using data for any analytics, simulation, and visualization practices.
The following sub-sections review each G2F dimension:

### 2.1. Dimension 1: G2F-Genomic Data (G2F-G)

The G2F has generated, stored, and released molecular genetic sequences at the level of single nucleotide polymorphism
(SNPs) for 1,576 lines tested across the environments. The SNPs are the most common type of genetic variation among
individuals. This data has been generated by a genotyping-by-sequence method known as GBS (McFarland et al., 2020). The
hierarchical data format (HDF) stores the sequenced raw SNPs data of all tested cultivars for data reliability and storage
efficiency. The raw genomic data stored in one single HDF file is available through G2F platform for public access. Figure 2
shows a screenshot of a slice of G2F-G hierarchical database stored in a single HDF file.





```
Keys: ['Genotypes', 'Positions', 'Taxa', '__DATA_TYPES__']

Genotype Lenght: ValuesViewHDF5(<HDF5 group "/Genotypes" (1579 members)>)

Shape:

<HDF5 dataset "AncestralAlleles": shape (945574,), type "<i4">
<HDF5 dataset "ChromosomeIndices": shape (945574,), type "<i4">
<HDF5 dataset "Chromosomes": shape (10,), type "|O">
<HDF5 dataset "Positions": shape (945574,), type "<i4">
<HDF5 dataset "ReferenceAlleles": shape (945574,), type "<i4">
<HDF5 dataset "SnpIds": shape (945574,), type "|S15">

Genotypes Data:

(CML442-B
(LAMA2002-23-3-B
(LAMA2002-35-2-B-B-B-B
(TX736)_((TX772_X_T246)_X_TX772)-1-5-B-B-B-B-B-B6-B6-B2-B13:100000550
(TX739)_LAMA2002-10-1-B-B-B-B3-B7_ORANGE-B:100000510
(Tx736)((Tx772xT246)xTx772)-1-5-B-B-B-B-B-B6-B12-B2-B13:100000968
(Tx739)LAMA2002-10-1-B-B-B3-B7orange-B7-B11:100000969
2FACC:100000938
2FACC:100001100
2MCDB:100000307
2MCDB:100000475
3IIH6:100000120
4N506:100000586
511811-1-1-B:100000114
511815-1-1-B:100000115
511828-1-1-B:100000142
511837-1-1-B:100000136
511842-1-1-B:100000119
511865-1-1-B:100000117
```

**Figure 2. A screenshot of the raw G2F-G data stored in a single HDF file showing complex hierarchical data structure of SNPs sequences.**


The published G2F-G HDF file is designed to be processed by the software Trait Analysis by aSSociation, Evolution and Linkage (TASSEL). TASSEL contains statistical approaches for trait association mapping, evolutionary patterns, and disequilibrium linkage (tasselsoftware.com, Bradbury et al., 2007). Figure **3** is a screenshot of a portion of the G2F molecular markers dataset open in TASSEL, illustrating comprehensive structure of genetic sequences.




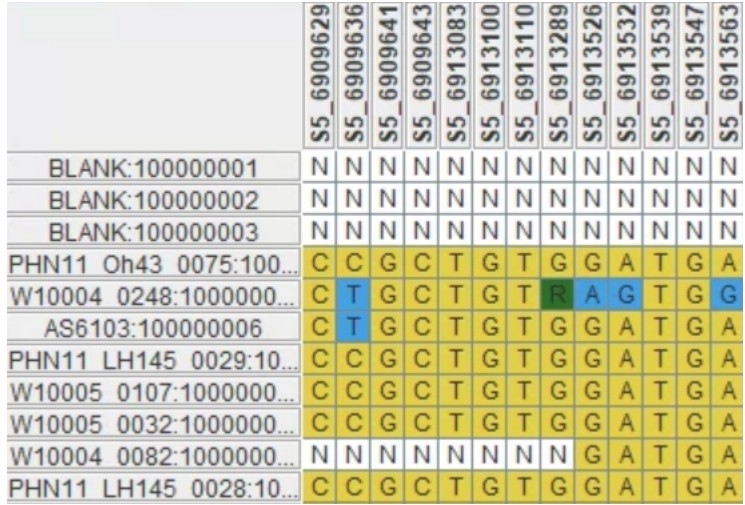

**Figure 3.** A screenshot of the raw G2F-G molecular markers sequences data stored in a single HDF file in TASSEL software. The first column shows the maize hybrid genotype names, and the first row shows the locus stored in the HDF file. The A, T, G, C, and R letters are a sample of the major and minor alleles at each molecular site, and N letter denotes the missing markers in a genetic sequence.

## 2.2. Dimension 2: G2F-Maize Phenotypic Data (G2F-P)

Different types of phenotypic variables have been collected as part of the G2F experiment: time related traits recorded during the growing season such as number of days to silking or pollen or flowering traits; yield components such as plant height [cm],

ear height [cm], ear width [cm], and ear length [cm]; and harvest or end traits such as grain yield. Other traits like root or stalk lodging occurrence are monitored before the harvest, and the number of stands, grain moisture [%], and grain yield [bu A$^{-1}$] are collected at harvest. More additional information, phenotypic variables definition, and the measurement techniques and devices can be found in the Genomes to Fields Phenotyping Handbook (genomes2fields.org). All the mentioned variables for all cultivars are recorded and released annually in comma-separated values (.csv) format through the G2F platform. Figure 4

represents data types of different variables and shows a slice of the G2F-P dataset.

| | A | B | C | D | Z | AA | AB | AC | AD | AE |
|---|---|---|---|---|---|---|---|---|---|---|
| 1 | Year | Field-Location | RecId | Source | Plant Height [cm] | Ear Height [cm] | Stand Count [plants] | Root Lodging [plants] | Stalk Lodging [plants] | Grain Moisture [%] |
| 2 | 2014 | TXH1 | 2218825 | LOCAL_CHECK | 193 | 94 | 92 | | | 11.8 |
| 3 | 2014 | MNH1 | 2235804 | 13WJWE:CG102:1227 | 190 | 86 | 92 | 0 | 0 | 30.5 |
| 4 | 2014 | TXH1 | 2218560 | WE13-80ISO-227-X-POL-80 | 211 | 127 | 89 | | | 11.7 |
| 5 | 2014 | TXH1 | 2218682 | 13SAJL:NURSE:0145 | 196 | 91 | 88 | | | 12.4 |
| 6 | 2014 | TXH1 | 2218600 | WE13-195ISO-149-X-POL-195 | 213 | 107 | 87 | | | 12.9 |
| 7 | 2014 | TXH1 | 2218781 | WE13-80ISO-418-X-POL-80 | 211 | 97 | 87 | | | 12.2 |
| 8 | 2014 | IAH1a | 2185067 | 13WJWE:LH198:3022 | 227 | 120 | 87 | 0 | 1 | 21.2 |
| 9 | 2014 | TXH1 | 2218789 | WE13-80ISO-062-X-POL-80 | 211 | 127 | 87 | | | 12.1 |
| 10 | 2014 | TXH1 | 2218749 | WE13-80ISO-200-X-POL-80.2 | 175 | 84 | 87 | | | 11.8 |
| 11 | 2014 | TXH1 | 2218584 | LOCAL_CHECK | 188 | 99 | 86 | | | 12.2 |
| 12 | 2014 | TXH1 | 2218917 | WE13-195ISO-329-X-POL-195 | 196 | 107 | 86 | | | 12.1 |
| 13 | 2014 | TXH1 | 2218640 | WE13-195ISO-249-X-POL-195 | 229 | 114 | 86 | | | 11.8 |
| 14 | 2014 | TXH1 | 2218763 | WE13-80ISO-411-X-POL-80.2 | 193 | 86 | 86 | | | 11.2 |
| 15 | 2014 | TXH1 | 2218860 | WE13-195ISO-149-X-POL-195 | 218 | 137 | 86 | | | 11.9 |



**Figure 4. A screenshot of the raw G2F-P data stored in ".csv" file showing a complex database structure of phenotypic observations in 2014. The "Year" column shows the year of the G2F experiment, "Field-Location" column shows the shows the 4-character name of G2F experiment consisting of the state abbreviation in the two first characters and the name of the hybrid experiment in the last two characters tested in that state, the "Recid" column shows the ID of the phenotypic record, the "Source" column shows the source of the collected phenotypic sample portal, the "Plant Height [cm]" column shows the height of the plant in [cm], the "Ear height [cm]" column shows the height of the ear in [cm], the "Stand Count [plants]" column shows the number of plants per plot at harvest, the "Root Lodging [plants]" column shows the number of plants that show the root lodging per plot, the "Stalk Lodging [plants]"column shows the number of broken plants per plot at harvest, and the "Grain Moisture [%]" column shows the percentage of the water content in plant at harvest. The other phenotypic variables have been measured and stored in similar columns. The blank cells represent the missing values of phenotypic observations.**

## 2.3. Dimension 3: G2F-Environmental Data (G2F-E)

Each G2F trial field is equipped with a WatchDog 2700 weather station (genomes2fields.org). These weather stations record the environmental data, mainly the climatic drivers in maize growth during the growing season including temperature [T (°C)], dew point [DP (°C)], relative humidity [RH (%)], solar radiation [SR (W m$^{-2}$)], rainfall [R (mm)], wind speed [WS (m s$^{-1}$)],

wind direction [WD (degrees)], and wind gust [WG (m s$^{-1}$)]. The annual environmental data is collected using weather station at each experimental field with temporal resolution of 30 minutes and stored in comma separated values (.csv) format. Data collected from every weather station is stored in one file for each year and is accessible through G2F website. The nearest National Weather Station (NWS) in ASOS network to each of the G2F weather station installed in the trial field has been used for false data calibration by G2F collaborators across the G2F layout (Alkhalifah et al., 2018; Jarquin et al., 2021). The

hydroclimatic time series extracted from the NWS stations have been released along with the G2F hydroclimatic time series observed in the experiments. Figure 5 represents a screenshot of a slice of G2F-E data in 2014 data stored in ".csv" format.

| | A | B | C | D | E | F | G | H | I | J | K | L |
|---|---|---|---|---|---|---|---|---|---|---|---|---|
| 1 | Record Number | Experiment | Station ID | NWS Network | NWS Station | Day [Local] | Month [Local] | Year [Local] | Day of Year [Local] | Time [Local] | Datetime [UTC] | Temperature [C] |
| 2 | 1 | DEH1 | 9079 | DE_ASOS | GED | 9 | 5 | 2014 | 129 | 15:00:00 | 5/9/2014 19:00 | 23.06 |
| 3 | 2 | DEH1 | 9079 | DE_ASOS | GED | 9 | 5 | 2014 | 129 | 15:30:00 | 5/9/2014 19:30 | 23.22 |
| 4 | 3 | DEH1 | 9079 | DE_ASOS | GED | 9 | 5 | 2014 | 129 | 16:00:00 | 5/9/2014 20:00 | 22.44 |
| 5 | 4 | DEH1 | 9079 | DE_ASOS | GED | 9 | 5 | 2014 | 129 | 16:30:00 | 5/9/2014 20:30 | 22.94 |
| 6 | 5 | DEH1 | 9079 | DE_ASOS | GED | 9 | 5 | 2014 | 129 | 17:00:00 | 5/9/2014 21:00 | 22 |
| 7 | 6 | DEH1 | 9079 | DE_ASOS | GED | 9 | 5 | 2014 | 129 | 17:30:00 | 5/9/2014 21:30 | 21.39 |
| 8 | 7 | DEH1 | 9079 | DE_ASOS | GED | 9 | 5 | 2014 | 129 | 18:00:00 | 5/9/2014 22:00 | 20.56 |
| 9 | 8 | DEH1 | 9079 | DE_ASOS | GED | 9 | 5 | 2014 | 129 | 18:30:00 | 5/9/2014 22:30 | 20.22 |
| 10 | 9 | DEH1 | 9079 | DE_ASOS | GED | 9 | 5 | 2014 | 129 | 19:00:00 | 5/9/2014 23:00 | 19.89 |
| 11 | 10 | DEH1 | 9079 | DE_ASOS | GED | 9 | 5 | 2014 | 129 | 19:30:00 | 5/9/2014 23:30 | 19.17 |
| 12 | 11 | DEH1 | 9079 | DE_ASOS | GED | 9 | 5 | 2014 | 129 | 20:00:00 | 5/10/2014 0:00 | 18.11 |
| 13 | 12 | DEH1 | 9079 | DE_ASOS | GED | 9 | 5 | 2014 | 129 | 20:30:00 | 5/10/2014 0:30 | 17.17 |
| 14 | 13 | DEH1 | 9079 | DE_ASOS | GED | 9 | 5 | 2014 | 129 | 21:00:00 | 5/10/2014 1:00 | 16.83 |
| 15 | 14 | DEH1 | 9079 | DE_ASOS | GED | 9 | 5 | 2014 | 129 | 21:30:00 | 5/10/2014 1:30 | 16.39 |
| 16 | 15 | DEH1 | 9079 | DE_ASOS | GED | 9 | 5 | 2014 | 129 | 22:00:00 | 5/10/2014 2:00 | 16.22 |
| 17 | 16 | DEH1 | 9079 | DE_ASOS | GED | 9 | 5 | 2014 | 129 | 22:30:00 | 5/10/2014 2:30 | 16.28 |
| 18 | 17 | DEH1 | 9079 | DE_ASOS | GED | 9 | 5 | 2014 | 129 | 23:00:00 | 5/10/2014 3:00 | 16.11 |
| 19 | 18 | DEH1 | 9079 | DE_ASOS | GED | 9 | 5 | 2014 | 129 | 23:30:00 | 5/10/2014 3:30 | 16.22 |
| 20 | 19 | DEH1 | 9079 | DE_ASOS | GED | 10 | 5 | 2014 | 130 | 0:00:00 | 5/10/2014 4:00 | 16.78 |
| 21 | 20 | DEH1 | 9079 | DE_ASOS | GED | 10 | 5 | 2014 | 130 | 0:30:00 | 5/10/2014 4:30 | 17.28 |
| 22 | 21 | DEH1 | 9079 | DE_ASOS | GED | 10 | 5 | 2014 | 130 | 1:00:00 | 5/10/2014 5:00 | 17.67 |
| 23 | 22 | DEH1 | 9079 | DE_ASOS | GED | 10 | 5 | 2014 | 130 | 1:30:00 | 5/10/2014 5:30 | 17.94 |
| 24 | 23 | DEH1 | 9079 | DE_ASOS | GED | 10 | 5 | 2014 | 130 | 2:00:00 | 5/10/2014 6:00 | 18.17 |
| 25 | 24 | DEH1 | 9079 | DE_ASOS | GED | 10 | 5 | 2014 | 130 | 2:30:00 | 5/10/2014 6:30 | 19 |
| 26 | 25 | DEH1 | 9079 | DE_ASOS | GED | 10 | 5 | 2014 | 130 | 3:00:00 | 5/10/2014 7:00 | 19.44 |
| 27 | 26 | DEH1 | 9079 | DE_ASOS | GED | 10 | 5 | 2014 | 130 | 3:30:00 | 5/10/2014 7:30 | 19.5 |
| 28 | 27 | DEH1 | 9079 | DE_ASOS | GED | 10 | 5 | 2014 | 130 | 4:00:00 | 5/10/2014 8:00 | 20.28 |
| 29 | 28 | DEH1 | 9079 | DE_ASOS | GED | 10 | 5 | 2014 | 130 | 4:30:00 | 5/10/2014 8:30 | 20.67 |
| 30 | 29 | DEH1 | 9079 | DE_ASOS | GED | 10 | 5 | 2014 | 130 | 5:00:00 | 5/10/2014 9:00 | 20.78 |
| 31 | 30 | DEH1 | 9079 | DE_ASOS | GED | 10 | 5 | 2014 | 130 | 5:30:00 | 5/10/2014 9:30 | 21.17 |
| 32 | 31 | DEH1 | 9079 | DE_ASOS | GED | 10 | 5 | 2014 | 130 | 6:00:00 | 5/10/2014 10:00 | 21 |
| 33 | 32 | DEH1 | 9079 | DE_ASOS | GED | 10 | 5 | 2014 | 130 | 6:30:00 | 5/10/2014 10:30 | 20.94 |
| 34 | 33 | DEH1 | 9079 | DE_ASOS | GED | 10 | 5 | 2014 | 130 | 7:00:00 | 5/10/2014 11:00 | 20.94 |





**Figure 5. A screenshot of the raw G2F-E data stored in ".csv" file showing a complex database structure environmental time series in 2014. The "Record Number" column shows the number of weather station records in each experiment, the "Experiment" column shows the 4-character name of G2F experiment consisting of the state abbreviation in the two first characters and the name of the hybrid experiment in the last two characters tested in that state, the "Station ID" column shows the ID of the weather station, "NWS Network" and "NWS Station" columns show the nearest NWS network and station has been used for initial QC by the G2F collaborators, the "Day [Local]", "Month [Local]", "Year [Local]", and "Day of Year [Local]" columns show the local day, month, year, and day of year of the weather record, "Daytime [UTC]" column shows the coordinated universal time, and the "Temperature [C]" column shows the temperature time series in [C]. The other climatic time series are collected and stored in similar columns.**

### 2.3.1. External environmental databases

To gap-fill the climatic datasets, we need to use externally accessible databases. Here three publicly available databases are proposed to use for this purpose: (1) National Solar Radiation Database (NSRDB); modeling and integrating half-hourly 4×4 km$^2$ meteorological dataset in the nation developed by the U.S. Department of Energy (Sengupta et al., 2018), (2) DayMet; 1×1 km$^2$ Daily Surface Weather and Climatological Summaries developed by Thornton et al. (Thornton et al., 2018), and (3) The Automated Surface Observing Systems (ASOS); developed by National Weather Service (NWS) which is a station-based program containing daily and sub-daily historical and forecasting hydroclimate. These public databases release temperature (°C), dew point (°C), relative humidity (%), solar radiation (W m$^{-2}$), rainfall (mm), pressure (mb and Pa), wind speed (m s$^{-1}$), wind direction (degrees), and precipitable water (mm).

### 2.4. Dimension 4: G2F-Metadata (G2F-M)

The metadata information is supplementary data about each experiment, including the name, ID, year, state, city, farm name, planting and harvesting dates, weather station serial number, weather station geo-location, and farm boundaries. These metafiles are released annually in comma-separated values (.csv) format through the G2F website. Figure 6 represents a screenshot of a slice of G2F-M data in 2014 stored in ".csv" format.

| | A | B | C | D | E | F | G | H |
|---|---|---|---|---|---|---|---|---|
| 1 | Location name | Type | Experiment | City | Farm | Field | lon | lat |
| 2 | DE | hybrid | DEH1 | Georgetown | Elbert N. & Ann V. Carvel Research & Education Center | 27AB | -75.204048 | 38.637405 |
| 3 | GA | hybrid | GAH1 | Tifton | Bellflower | 18 | -83.555016 | 31.506544 |
| 4 | IA1 | hybrid | IAH1 | Ames | Worle | | -93.696188 | 41.99653 |
| 5 | IA2 | hybrid | IAH2 | Carroll | | | -94.727606 | 42.066206 |
| 6 | IA3 | hybrid | IAH3 | Keystone | | | -92.259751 | 41.98713 |
| 7 | IA4 | hybrid | IAH4 | Crawfordsville | Southeast Research Farm | 14 | -91.486943 | 41.198645 |
| 8 | IL1 | hybrid | ILH1 | Urbana | Maxwell Farms | MF500 | -88.233184 | 40.061135 |
| 9 | IN | hybrid | INH1 | West Lafayette | Purdue ACRE | 97/98 | -87.006 | 40.488 |
| 10 | MN | hybrid | MNH1 | Waseca | Southern Research & Outreach Center | NA | -93.53409096 | 44.06971707 |
| 11 | MO1 | hybrid | MOH1 | Columbia | Bradford | C1a | -92.20997 | 38.898702 |
| 12 | MO2 | hybrid | MOH2 | Columbia | Rollins=Hinkson Creek Bottoms | block 5 | -92.352163 | 38.928745 |
| 13 | NC | hybrid | NCH1 | Kinston | Cunningham Research Farm | L block 5 | -77.57159444 | 35.29921111 |
| 14 | NE1 | hybrid | NEH1 | Lincoln | East Campus | 1807 | -96.656687 | 40.834392 |
| 15 | NE2 | hybrid | NEH2 | North Platte | Dryland farm | | -100.749467 | 41.052978 |
| 16 | NE3 | hybrid | NEH3 | Brule | North Dryland | West 1/4 | -101.99598 | 41.16353 |
| 17 | NY1 | hybrid | NYH1 | Aurora | Musgrave Research Farm | J | -76.651654 | 42.728765 |
| 18 | NY2 | hybrid | NYH2 | Aurora | Musgrave | E4 | -76.65 | 42.73 |
| 19 | ON1 | hybrid | ONH1 | Waterloo | Rosdendale | Huras | -80.42696111 | 43.497025 |
| 20 | ON2 | hybrid | ONH2 | Ridgetown | On Campus | Range 5 | -81.88311111 | 42.45419722 |
| 21 | TX1 | hybrid | TXH1 | College Station | University Farm | 224 | -96.43394444 | 30.54684444 |
| 22 | TX2 | hybrid | TXH2 | halfway | Halfway | pivot | -101.9494444 | 34.18466667 |
| 23 | WI | hybrid | WIH1 | Madison | West Madison | M1400 | -89.53098611 | 43.05706111 |

**Figure 6. A screenshot of the raw G2F-M data stored in ".csv" file showing a complex data structure metadata in 2014. The "Location Name" column shows the state and the number of the experiment in that state, the**





**"Type" column shows the type of the experiment which can be hybrid or inbred, the "Experiment" column shows the 4-character name of G2F experiment consisting of the state abbreviation in the two first characters and the name of the hybrid experiment in the last two characters tested in that state, the "City" column shows the city that the experiment located at, the "Farm" column shows the name of the farm that the experiment has been tested in, the "Field" column shows the name of the field of the experiment, and "lon" and "lat" columns show the longitude and the latitude of the weather station installed in the field.**

## 3. Methodology

### 3.1 Database quality control

The QC-CC is a two-module data preprocessing pipeline developed in Python for each of the G2F data dimensions (G2F-G, G2F-P, G2F-E, and G2F-M) released between 2014 and 2017 (Fig. 7). The QC module focused on four general phases, and

they have specific extensions for each data dimension. The general QC phases are:

1) Reading raw files.

2) Checking the data format and structure.

3) Detection of missing values and data gaps in the datasets, and

4) Implementation of predictive data analytics to fulfill gaps.

In the first step, the raw files for G2F-P, G2F-E, and G2F-M are read to identify whether the necessary information is recorded in the right column with the appropriate header name (some headers are presented in Fig. 4-6). The complete lists of appropriate headers for each data dimension are represented in Sect. 3.1.2-3.1.4. When the released files lack structure and consistent format, the next step is to correct the respective columns and header names. Then, the missing values in each dataset are searched and identified, and the appropriate QC methods (i.e., assign an average value for G2F-G and a predicted value based

on deep neural network for G2F-E; Sarzaeim et al., 2022a) are adopted to impute the missing values. After performing all steps above for each dataset, the quality-controlled datasets are restored in the updated files and transferred to the CC module. The subsections below, explain the methodological QC steps for each G2F data dimension (Fig. 7 illustrates the associated algorithm).

### 3.1.1 Sub-Module 1: G2F-G

The G2F stores and releases genomic sequences data in HDF file. It is noteworthy that unlike the phenotypic, environmental, and metadata been annually released through the G2F website, the genomic data file has been made available once in a consolidated HDF file containing the molecular marker sequences of all maize inbred lines used as parents of the hybrids tested in all G2F experiments.

First, we downloaded the raw genotypic data file from the G2F platform, converted to text (.txt) format, named "Markers.txt"

and saved in "File Upload/Genotype" directory in the database package (Sarzaeim et al., 2023). The text file is then preprocessed to (1) convert the SNPs to numerical genotypic data, (2) exclude the genotypes with large percent of missing

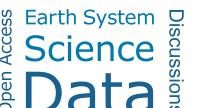

values in their genetic sequence, (3) exclude the genotypes that lack of allelic variation, and (4) impute the missing SNPs for the remaining cultivars (see Fig. 7). These steps were integrated and implemented in a single script in Python named "01_Transformations.py" located at "G2F data preprocessing/Genotype" directory as follows:

1) The raw HDF file released by G2F has been created in the structure that works only in the TASSEL as a "black box" software. The developed script extracts the molecular genetic markers from the text file and converts them to numerical genotypes in csv-format. This step facilitates the processing of the SNPs within the Python environment. The numerical genotype values are the probability of a major allele to be selected randomly in a site marker. Thus, the minor and major allele homozygous are converted to 0 and 1, respectively; and the heterozygous are converted to

0.5.

2) A script was developed to discard the cultivars with more than 20% missing values in their genetic sequence, providing enough DNA information for further analyses. The 20% threshold percentage is called the percent of missing values (PMV), which varies according to the criteria of the data user. Here, we used the PMV proposed by Jarquín et al. (2017).

3) The SNPs with a minor allele frequency (MAF) smaller than 3% were removed. This filter aims to discard the genotypes that lack allelic variation. As in the previous step, the MAF threshold used by Jarquin et al. (2017).

4) The remaining missing SNPs for each individual are fulfilled using the average of the numerical genotypes at each locus (p). If the average is equal to or smaller than 0.5 (the probability of heterozygous selection), the missing values are fulfilled by the p. Otherwise, the missing values are imputed by 1-p. The screened lines and their fulfilled SNPs

sequences are generated and stored in a clean version of genotypic data in ".csv" format.

### 3.1.2. Sub-Module 2: G2F-P

Multiple participants affiliated to the G2F initiative monitored Maize's growth stages and harvest (genomes2fields.org). Examples of phenotypes include plant morphology (e.g., plant height [cm]), ear morphology (e.g., ear height [cm], width [cm], and length [cm]), and plant productivity (e.g., grain moisture [%] and yield [bu A$^{-1}$]). While in this study we focused on yield

for simulation and prediction purposes, measured in [bu A$^{-1}$], other phenotypes are made available and can be used.

The phenotypic datasets are released on an annual basis through the G2F website in ".csv" format. First, for preprocessing, we download the raw data files from all available years, save them in "File Upload/Phenotype" directory and then the QC is implemented to (1) check whether the first-level data known as primary columns are available, (2) check whether the second-level data known as secondary columns are available, and (3) remove the missing samples (Fig. 7). These steps are described

below:

1) The primary columns are the first-level data necessary for further processing. These columns are "Year," "Field-Location," "Pedigree," "Plant Height [cm]," "Ear Height [cm]," "Grain Moisture [%]," and "Grain Yield [bu A$^{-1}$]." The Python script "01_Phenotype_Files_Primary_Columns.py" verifies if the mentioned headers are available in the phenotypic files. Note that the input is case-sensitive, and in many cases, there are typos in headers in the raw files.



Thus, the script returns the associated error(s) with typos and suggests how to fix them. The user fixes those typos manually in the raw files. Otherwise, the file is ready for the secondary-column control step.

2) The secondary columns represent the second-level data necessary for further analysis, but if they are not available in the raw files, they can be constructed based on primary columns. These columns are "ID," "Experiment," "Experiment ID," "Pedigree," "P1," and "P2." The "Location" denotes the state and the name of the hybrid experiment. The
"Experiment" refers to the environment, year, state, and name of the hybrid experiment. The "Experiment ID" refers to the unique ID, which is the combination of the hybrid experiment's year, state, and name. The "P1" and "P2" denote the maize hybrid parental pedigrees' names. The Python script "02_Phenotype_Files_Secondary_Column.py" controls the availability of these columns. If they are not available in the raw files, they will be created automatically from the data available in the primary columns.

3) We need the phenotypic observations to train and test the crop growth model (e.g., GxE model). In many cases, the phenotype's observed measurements have been missed to be recorded, and thus, the missing phenotypic samples are filtered out from the database by applying "01_Phenotypes.py" script.

The developed Python scripts for step (1) and (2) are located at "File Control/Phenotype" directory, and the script for step (3) is located at "G2F data pre-processing/Phenotype" in the database package.

### 3.1.3. Sub-Module 3: G2F-E


The G2F environmental time series consists of temperature [T (°C)], dew point [DP (°C)], relative humidity [RH (%)], solar radiation [SR (W m$^{-2}$)], rainfall [R (mm)], wind speed [WS (m s$^{-1}$)], wind direction [WD (degrees)], and wind gust [WG (m s$^{-1}$)] collected during the growing season, from planting to the harvest. The following QC steps and the developed Python scripts are designed to preprocess the above hydroclimatic variables. The users can adapt the scripts to integrate other environmental
time series.

G2F-P and G2F-E QC steps are similar except for some extensions of the latter. The G2F-P datasets are single measurements sampled at a specific maize growing stage for each individual plant, while the G2F-E datasets are time series of continuous hydroclimate records along the maize growing season for each experimental site. The hydroclimate time series data required additional pre-processing actions to form the G2F-E QC. The additional actions include the initial elimination of erroneous
hydroclimatic records, corrections of experiment name, and dataset categorizations accounting for the missing values.

For G2F-E preprocessing, we first download the raw data files from all available years; then, we save the data files in "File Upload/Environment" directory in the database package and implement the QC. The QC procedure (1) checks whether the first-level data, known as primary columns, are available, (2) checks whether the second-level data known as secondary columns are available, (3) checks whether the missing samples in each experiment in each year are existing, and (4) imputes
the data gaps (see Fig. 7). These steps are described below in detail:

1) The primary columns are the first-level data necessary for further processing. These columns are "Station ID," "Experiment," "Day [Local]," "Month [Local]," "Year [Local]," "Time [Local]," "Temperature [C]," "Dew Point [C],"





“Relative Humidity [%],” “Solar Radiation [W m$^{-2}$]”, “Rainfall [mm],” “Wind Speed [m s$^{-1}$],” “Wind Direction [degrees],” and “Wind Gust [m s$^{-1}$].” The Python script “01_Weather_Files_Primaty_Column.py” located in subdirectory “File Control/Environment” checks if these columns exactly with the mentioned headers are available in the environmental files. Note that, like the G2F-P, the input is case-sensitive. Thus, the script exactly returns the associated error where there is a mismatch and provides suggestions for fixing typos. Also, the user needs to fix the typos manually in the raw files, otherwise the file is ready for the next control step.

2)  The secondary columns are the second-level data necessary for further analysis, but if they are not available in the raw files. The columns for weather data are “Record Number” and “Day of Year [Local]”. The Python script “02_Weather_Files_Secondary_Column.py” located in “File Control/Environment” controls the availability of these columns. If the columns are not available in the raw files, they will be created automatically from the data available in the primary columns.

3)  Before checking for the missing values, we can perform an initial check on the time series and remove the remained erroneous samples after the G2F collaborators implemented the QC. The script “03_Control.py” is saved in the “File Control/Environment” directory. This initial check occurs in the Python script and depends on the weather variables and their possible value range:

•  For “Relative Humidity [%]” the script removes the $x$ values if $x < 0$ or $x > 100$.

•  For “Solar Radiation [W m$^{-2}$]” the script removes the $x$ values if $x < 0$.

•  For “Rainfall [mm]” the script removes the $x$ values if $x < 0$.

•  For “Wind Direction [degrees]” the script removes the $x$ values if $x < 0$ or $x > 360$; and assigns an $x$-value to empty if the “Wind Speed [m s$^{-1}$]” is zero

For further analysis, we need to have a consistent and informative protocol for uniquely name the experiments because of the multiple experiments implemented in each state and field. Additionally, the name's format should be consistent in the entire QC module. We created a name format that illustrates the split of the raw files into as many “.csv” files as experiments are recorded in each raw environmental file. The newly-generated file names are self-described as “YearStateExperiment”. For example, “2014ILH1.csv” refers to the environmental file containing the weather time series recorded for experiment “H1” implemented in the state of “IL” in the year “2014” and stored in “.csv” format. The scripts, “01_Weather_Data_Reading.py” that reads the environmental data with correct primary and secondary columns and correct the values from all years, and “02_Name_Fixing.py” that fixes the experiments names, both are in the “G2F data preprocessing/Environment” directory.

The environmental datasets are categorized into three groups based on the presence of missing values in the raw environmental data files: (1) “complete,” (2) “empty,” and (3) “incomplete.” The separate Python scripts “Database.py” for each hydroclimate variable go through the generated files with a specific name containing the environmental time series for each experiment in each year to check if all the records during the growing season are available or not. For example, if all records of temperature for a given experiment are available, this dataset belongs



to the "complete" group. If all temperature records are empty, that dataset belongs to the "empty" category. If the temperature dataset is not categorized into the above groups, it belongs to "incomplete" category. The "complete" datasets are directly transferred to the updated environmental database ready for CC module. However, the "empty" and "incomplete" datasets must be imputed and fulfilled, and then moved to the improved database. A separate Python script has been developed to categorize each hydroclimatic variable into the three groups above and within the "Database" subdirectory of the database package.

4) For gap fulfillment of "empty" and "incomplete" time series, we developed an evaluation-improvement pipeline (Sarzaeim et al., 2022a). This pipeline acquires external hydroclimate (i.e., NSRDB, DayMet, and NWS) through developed Application Programming Interfaces (APIs). The Python APIs are located at "API" folder in the database package for download, store, and process the G2F hydroclimate time series at the available locations and years. Afterwards, the script imputes the best-fitted dataset from the NSRDB, DayMet, or NWS for any given hydroclimate variable to the "empty" datasets. Following the data available at http://doi.org/10.5281/zenodo.7490246 (Sarzaeim, et al., 2023), the "incomplete" datasets use a separate script for predictive analytics of deep neural networks to cover the missing hydroclimate values in the G2F-E time series, which are stored in "ML" folder and are part of the database package. The updated "empty" and "incomplete" datasets are transferred to the updated improved G2F-E database, and later used by the CC module. For the ease of selecting the desired experiment(s) by users, a Python script has been developed and stored in the "Selection" folder of the database package and offers experiment options for users to select.

### 3.1.4. Sub-Module 4: G2F-M

The metadata files contain the digital information relevant to the experiments annually released at the G2F website in a ".csv" format. For preprocessing, we download the raw data files from all available years, save them in "File Upload/Meta" directory, and then implement the control. Then, the control (1) checks whether the first-level data known as primary columns are available, (2) checks whether the second-level data known as secondary columns are available, and (3) checks whether any experiments with unknown locations are available (see Fig. 7). The scripts for steps (1) and (2) are stored in "File Control/Meta" directory and the script designated for step (3) is located at "G2F data preprocessing/Meta" directory, all within the database package. These steps are described below in detail:

1) The primary columns are the first-level data necessary for further processing. These columns are "Experiment," "Lat," and "Lon". The "Lat" and "Lon" denote the latitude and longitude of the weather stations located in the field. The script "01_Meta_Files_Primary_Columns.py" first checks if these primary columns with the exactly listed headers are available in the metadata files. Note that the input is case-sensitive. Thus, the script returns the associated error where there is a mismatch and suggests how to fix them. In this case, the user needs to fix the typos manually in the raw files. Otherwise, the file is ready for the next control step.



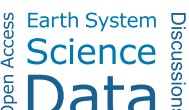

2)  The secondary columns are the second-level information necessary for further analyses. These columns are "State,"
"Experiment ID," and "Experiment Type". Note that there are two types of experiments conducted by the G2F
        collaborators: Inbred and Hybrid experiments. Here, we need the hybrid experiments for the GxE simulation. The
        script "02_Meta_Files_Secondary_Columns.py" controls the availability of secondary columns. If they are not
        available in the raw files, they will be created automatically from the information available in the primary columns.

3)  For model output postprocessing and geospatial visualization, the script "01_Lat_Lon_Reader.py" requires the
latitude and longitude of the experiments. Additionally, if a given dataset is categorized as "empty" or "incomplete,"
        the G2F experiment location is also required to geolocate and extract the associated values from other databases. The
        experiments with missing latitude and longitude are removed.

## 3.2 Consistency control

The CC module is the last pre-processing step before data is ready for model implementation (i.e., GxE modeling). The CC
module integrates all controlled and updated files from the QC module, checks their compatibility as inputs for GxE modeling,
and synthesizes the multi-dimensional database for phenotypic simulation and postprocessing. The compatibility check is
required by the GxE model, which is only possible when genomic, phenotypic, environmental data, and metadata are present.
When some genotypic markers or phenotypic observations or metadata are discarded in the QC sub-modules, the CC removes
the experiments with at least one missing dimension in the controlled files. The designed Python script for CC module is saved
in "Control" folder in the database package.

Figure 7 conceptualizes the QC-CC algorithm for each dimension. First, each dataset is controlled by its format, availability,
and imputation. Then, the quality-controlled datasets are evaluated for compatibility purposes for the simulation process in the
CC module.

The designed Python script for CC module is saved in "Control" folder in the database package.

Figure 7 conceptualizes the QC-CC algorithm for each dimension. First, each dataset is controlled by its format, data
availability, and imputation. Then, the quality-controlled datasets are evaluated for compatibility purposes for the simulation
process in CC module.





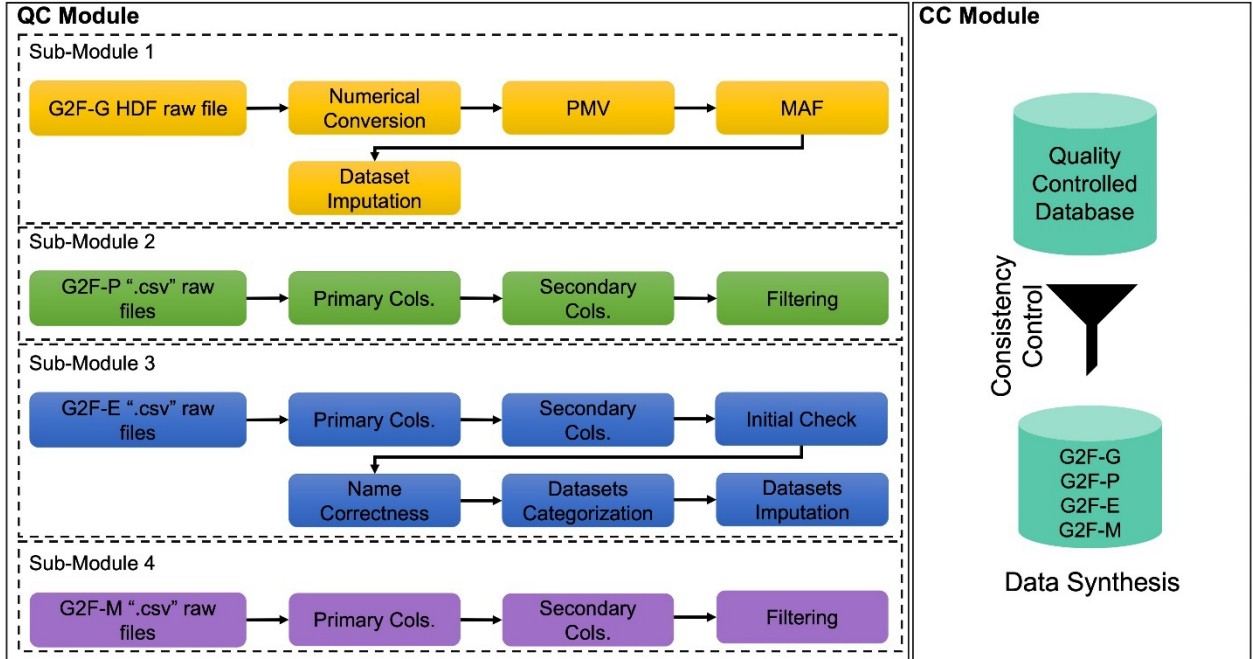

**Figure 7. The overall algorithmic QC-CC framework for G2F database. The "G2F-G," "G2F-P," "G2F-E," and "G2F-M" denote the G2F genomic, phenotypic, environmental, and meta data, respectively. The "PMV" and "MAF" denote the percent missing values and minor allele frequency, respectively. The "Primary Cols." and "Secondary Cols." denote primary and secondary columns, respectively.**

### 3.3. Uncertainty

For the quantification of uncertainty in improved climate data by other data sources (i.e., NSRDB, DayMet, and NWS) we
used the differences in the standard deviation (SD) between the climate time series of the G2F and other data sources used for
G2F-E data imputation. The SD statistics represent the dispersion of the probability distribution function (PDF) of errors and
measure the magnitude of the standard uncertainty according to Merchant et al. (2017). The following equation represents the
error term:

$$err_{G2F-option} = x_{m,t,G2F} - x_{m,t,option} \qquad option = NSRDB, DayMet, NWS \qquad (1)$$

Where, $err_{G2F-option}$ is the difference between G2F time series and other options, $x_{m,t,G2F}$ is the G2F observed value of
variable $m$ at day $t$, and $x_{m,t,option}$ is the value of variable $m$ from other options at day $t$. The uncertainty is estimated as a
spatial aggregate for the area of study. Yet, the algorithm can be implemented by station if the degrees of freedom is adequate.
A separate script "Uncertainty.py" was developed to quantify the uncertainty for each hydroclimatic variable located in
"Database" folder of the database package.



## 4. Results and discussion

In this study, we aim to introduce a quality and consistency data controls framework that includes the consolidation of pipelines for the retrieval, transformation, improvement, and access to spatiotemporal, large-scale, and multi-dimensional databases for plant breeding. The provided QC-CC pipeline uses a high-dimensional G2F database that involves genomic, phenotypic, environmental, and metadata, integrating and improving a database for maize yield predictability. The results of the QC module applications are presented in Sect. 4.1 to 4.4. The results of the CC module and data synthesis are presented in Sect. 4.5.

Finally, the uncertainty introduced by external environmental databases to improve the G2F-E is presented in Sect. 4.6.

### 4.1. G2F-G QC

Plant breeding and genetic improvement programs focus on developing more productive cultivars resistant to uncertain environmental conditions. These uncertain conditions include a wide range of biotic (i.e., diseases, pests, and herbicides) and abiotic (i.e., drought, heat, cold extremes, wet weather, and water limits) stresses (Blum, 2010) which directly affect the crops'

productivity and yields. The crop yield (and other commercially essential phenotypes) can be improved in the target environment by selecting the varieties tolerant to the environmental stresses (Cattivelli et al., 2008; Sarzaeim et al., 2021). The molecular markers data for tested lines in multiple environments across the large scale of the U.S. and Ontario in Canada provide the opportunity to diagnose and select superior and tolerant maize lines with specific environmental stresses in each environment.

There are extensively published datasets for phenotypic measurements, biophysical parameters, and geospatial environmental observations in croplands. Gomez-Dans et al. (2022) released an integrative dataset in West Africa, including location, leaf area index, and maize yield values. In another study, Weber et al. (2022) published a high-quality, multi-crop, and multi-year database during the crop phenological stages containing canopy height, leaf area index, biomass, and soil water content and temperature in Europe. However, the lack of genetic data may limit the ability to diagnose the superior lines. Thus, providing

and publishing high-quality crop genomic datasets and ground phenotypic and environmental observations adds value to designing climate-resilient cropping systems for a changing climate. Poland et al. (2012) and Jarquín et al. (2014) underscore that crop DNA data consist of missing values due to the technical inadequacy of sequencing. Also, Alkhalifah et al. (2018) described that the main limitations with G2F datasets, including the G2F-G, are missing data in several marker sites. We have previously observed the missing sequencing values in Fig. 3. To overcome this limitation, the generated numerical genotypes

for each maize line pass through the PMV to remove the genotypes containing missing values of more than 20% of the whole sequence. Along with PMV, the MAF filter eliminates the uncommon variants. Lopes et al. (2015) describe that rare variants are usually removed because of the limited population size and keeping the acceptable precision level in phenotyping.

After applying the PMV and MAF filters, 253 lines were removed, and 1,323 individuals with numerical genotypes were kept for further analysis. This process led to missing values in the genome sequences in the remanent cultivars of less than 20%,



and the minor allele frequency is larger than 3%. The defined strategy in Sect. 3.1.1 fulfills the missing values in marker sites of the remining 1,323 maize lines and the integrated, imputed, and enhanced G2F-G datasets are ready for further analysis.

### 4.2. G2F-P QC

Overall, phenotypic field measurements of 31,866 individual cultivars have been recorded for maize inbred and hybrid experiments between 2014 and 2017 across G2F sites. Figure 8 shows the spatial distribution of phenotypic measurements

sampled for each G2F experiment. The minimum and maximum observations are 93 and 807 sampled in the "2017GAH2" and "2014IAH1" experiments, respectively. The total number of observations from 2014 to 2017 was 5,834, 9,841, 7,524, and 7,175, respectively.

**Figure 8.** The spatial distribution of phenotypic records in G2F-P database in (a) 2014, (b) 2015, (c) 2016, and (d) 2017. The largest total number of measurements is in 2015 with 9,834 samples.

Like the G2F-G, several missing values are existing in the phenotypic observations like in Fig. 4. Note that the target phenotypic measurement is maize grain yield in this study; thus, the missing values for grain yield are removed from the raw

phenotype datasets. However, the same methodology is applicable for other phenotypic variables illustrated in Fig. 4, as well.

By removing cultivars with grain yield missing values, a total of 30,014 field observations remains in the G2F-P dataset. In the last step, the clean versions of G2F-P dataset in each year between 2014 and 2017 are consolidated in one single ".csv" file. One record of the clean G2F-P dataset is represented in Table 1 as an example. This example displays phenotypic observations for the B37/MO17 maize line tested in the state of Delaware in the experiment of H1 in 2014.

**Table 1. Record of a single of G2F-P dataset. It shows the phenotypic measurements including "Plant Height (cm)," "Ear Height (cm)," "Grain Moisture (%)," and "Grain Yield (bu A$^{-1}$)" for a maize hybrid with pedigrees of "B37" and "MO17" collected in "2014-DEH1" experiment located in Delaware in 2014. The ID of the record is "2014_DEH1_B37/MO17, and the ID of the experiment is "2014DEH1". The "H" denotes the hybrid type of the experiment, "P1" and "P2" denote the pedigrees of the maize hybrid, and "DE" denotes the state of Delaware.**

| ID | Year | Location | Experiment | Experiment ID | Pedigree | P1 | P2 | Plant Height (cm) | Ear Height (cm) | Grain Moisture (%) | Grain Yield (bu A$^{-1}$) |
|---|---|---|---|---|---|---|---|---|---|---|---|
| 2014_DEH1_B37/MO17 | 2014 | DEH1 | 2014-DEH1 | 2014DEH1 | B37/MO17 | B37 | MO17 | 235 | 139.5 | 19.2 | 217.2 |


### 4.3. G2F-E QC

The designed QC scripts in Python for hydroclimatic files have been implemented, and the available typos and mismatches in the headers have been fixed to have a consistent format among the files stored in different years.

The nonviable samples available in the datasets, such as negative values for solar radiation and rainfall, the out-of-range
relative humidity percentage, and the wrong wind direction values have been detected, eliminated, and left as missing values, as described in Sect. 3.1.3.

At this point, the naming policy for the environments is applied. Note that this study focuses on the hybrid experiments for GxE models and associated simulations, which suggests that inbred experiments are discarded. One hundred twelve hybrid experiments remain in the database for the categorization step.

The G2F-E QC and G2F-M QC sub-modules are implemented in parallel. The reason for this parallel implementation is: (1) the geolocation of weather stations is required to download the data from external environmental data sources, and (2) the location of the experiments is required for the visualization of the geospatially distributed crop growth predictability. Among the 112 experiments, there are 15 experiments with missing data. Afterwards, for simplicity of the datasets analyses, each G2F annual climate ".csv" file is split into separate files for each experiment and climate variable. This file structure represents
eight files containing each of the hydroclimatic variables time series (e.g., temperature, dew point, relative humidity, solar





radiation, rainfall, wind speed, wind direction, and wind gust) for each experiment (97×8 = 776 time series files are created and stored).

On the other hand, just 32 experiments were complete from the 97 experiments that compose the file structure. Table 2 presents a synthesis of experiment completeness between 2014 and 2017 for the G2F-E data. The missing files are mainly caused by gaps of environmental data, limiting the ability of crop models and analytics for phenotype predictions. This situation was emphasized by Huang et al. (2019) who evidenced that the limitation in phenotypic and environmental data restricts the timely diagnostics of crop growth and, consequently, hampers the use of crop growth models for prediction purposes. Di Paola et al., 2016 provides an additional perspective by using the minimal set of input data for crop growth modeling predictions can become more biased. Sarzaeim et al. (2022a) provided a strategy to reduce the gaps in environmental data using deep neural networks. Such effort evidenced how phenotype predictability increases and could be attributed to climate patterns of variability.

**Table 2. The percentage of complete, empty, and incomplete portions of time series for each G2F hydroclimatic variable: Temperature (T), Dew Point (DP), Relative Humidity (RH), Solar Radiation (SR), Rainfall (R), Wind Speed (WS), Wind Direction (WD), and Wind Gust (WG).**

|  | T (°C) | DP (°C) | RH (%) | SR (W m$^{-2}$) | R (mm) | WS (m s$^{-1}$) | WD (degrees) | WG (m s$^{-1}$) |
|---|---|---|---|---|---|---|---|---|
| Complete | 79 | 71 | 80 | 49 | 77 | 79 | 79 | 61 |
| Empty | 0 | 6 | 0 | 12 | 1 | 1 | 1 | 5 |
| Incomplete | 21 | 23 | 20 | 39 | 22 | 20 | 20 | 34 |

In this study, we fulfill the missing values identified as empty and incomplete in the environmental time series to consolidate a high dimensional database that could be translated into an improvement in GxE models performance. The improved G2F-E enhances the G2F multi-dimensional database and provides the opportunity to increase the OMICs observations engaged in the GxE simulations. The time series without missing values are delivered to the final improved database, while files with empty or incomplete time series are processed to fulfill data gaps with external climate data sources (e.g., NSRDB, DayMet, NWS). For fulfillment step, the designed APIs read the "Lat" and "Lon" data from controlled G2F metafiles, download, and store the climatic datasets for each G2F experiment trial site. The downloaded datasets for each data source are divided into separated files, one per experiment and climate variable, and stored in ".csv" format.

The empty datasets have been replaced by one of the other data sources selected based on the calculated minimum root mean square error (RMSE) values between G2F and each of the NSRDB, DayMet, and NWS for a given climatic variable in G2F database. A deep neural networks (DNNs) technique was implemented to estimate the missing values of the incomplete datasets. The strategies for gaps fulfillment have been explained in detail in Sarzaeim et al. (2020;2022a,b). The gap fulfilment in the environmental data allowed us to increase the number of complete experiments from 32 to 86 experiments. Also, we added other climatic variables like pressure and precipitable water from NSRDB and DayMet, which were not initially





provided by the G2F initiative. The G2F-E QC sub-module enables downloading other databases and pre-process them for the expansion of the G2F-E.

One record of the improved G2F-E data is represented in Table 3 as an example. This example refers to a record for a hybrid experiment called H1 conducted in the state of Delaware in 2014. This record represents the first observation of the climatic time series, including temperature, dew point, relative humidity, solar radiation, rainfall, wind speed, wind direction, and wind

gust.

**Table 3. Record of a single example of G2F-E dataset. It shows the observed hydroclimate data including "Temperature (°C)," "Dew Point (°C)," "Relative Humidity (%)," "Solar Radiation (W m⁻²)," "Rainfall (mm)," "Wind Speed (m s⁻¹)," "Wind Direction (degrees)," and "Wind Gust (m s⁻¹)" collected by weather station with ID of "9079" for "2014DEH1" experiment located in Delaware on 9 May 2014 at 15:00:00 local time. The ID of the experiment is "2014DEH1". The "H" denotes hybrid type of the experiment,**

**and "DE" denotes the state of Delaware.**

| Record Number | Station ID | Location | Experiment ID | Day [Local] | Month [Local] | Year [Local] | Day of Year [Local] | Time [Local] | Temperature (°C) | Dew Point (°C) | Relative Humidity (%) | Solar Radiation (W m⁻²) | Rainfall (mm) | Wind Speed (m s⁻¹) | Wind Direction (degrees) | Wind Gust (m s⁻¹) |
|---|---|---|---|---|---|---|---|---|---|---|---|---|---|---|---|---|
| 1 | 9079 | DEH1 | 2014DEH1 | 9 | 5 | 2014 | 129 | 15:00:00 | 23.06 | 15.78 | 63.2 | 887 | 0 | 1.79 | 32 | 4.02 |

### 4.4. G2F-M QC

From 2014 to 2017, a total of 112 tested hybrid experiments were registered across the G2F sites. However, the latitude and longitude of 15 experiments were missed and consequently removed from the database. As mentioned in Sect. 4.3, the G2F-M QC sub-module has been implemented in parallel with the G2F-E QC sub-module to avoid the processing of redundant data

for the experiments with unknown location. One record of the G2F-M data is represented in Table 4 as an example. This example illustrates the coordinates of the weather station located in the experiment of H1 in the state of Delaware in 2014.

**Table 4. Record of a single of G2F-M dataset. It shows the location including "Lat" and "Lon" of the "2014DEH1" experiment located in Delaware in 2014. The ID of the experiment is "2014DEH1". The "Lat" denotes latitude, "Lon" denotes longitude, "H" denotes the hybrid type of the experiment, and "DE" denotes the state of Delaware.**

| Experiment | Experiment ID | Experiment type | Year | State | Lat | Lon |
|---|---|---|---|---|---|---|
| DEH1 | 2014DEH1 | H | 2014 | DE | 38.63 | -75.20 |




## 4.5. Database CC

The last stage of input data preprocessing is to check the consistency among the quality-controlled and improved files across the G2F-G, G2F-P, G2F-E, and G2F-M QC sub-modules. The main purpose of the CC module is to check all quality-controlled files and remove those from the records when their information is not available. In other words, the CC module records the

available files with complete sequences of genetic, phenotypic observations, climatic time series, and location data for an eventual implementation of GxE model and visualization analytics, or the possible use in crop and Earth System models. Also, the CC uses the unique experiments' names in the "Experiment ID" column, which is common among G2F-P, G2F-E, and G2F-M, to remove those records missing at least one OMICs or environmental category of G2F data. After checking this three data dimensions consistency, the CC module uses "P1" and 'P2" columns, common between controlled G2F-P and G2F-G, to

update the G2F-G file for the available records in phenotypic data. Consequently, all the common records in the high dimensional G2F data are kept for use in crop growth modeling. We identified that after implementing the CC on G2F's 2014-2017, 376 lines, 8,171 yield observations, and 84 experiments remained for phenotype diagnostics or modeling. Figure 9 symbolizes the synthesis of enhanced high-dimensional G2F database after applying QC and CC modules.

The considerable decrease in the number of genotypes indicates that although the genetic sequence of 1,576 maize lines have

been generated and published in G2F database, most of them have not yet been tested in the trials. The phenotypic observations dropped from 31,886 to 8,171 after QC-CC, which could be mitigated by releasing the new samples in a larger number of experiments by G2F initiative through years and overcome this trials deficit. The use of crop and data-driven modeling, and remote sensing products to estimate the crop yield and other phenotypes can mitigate these data deficits as well.

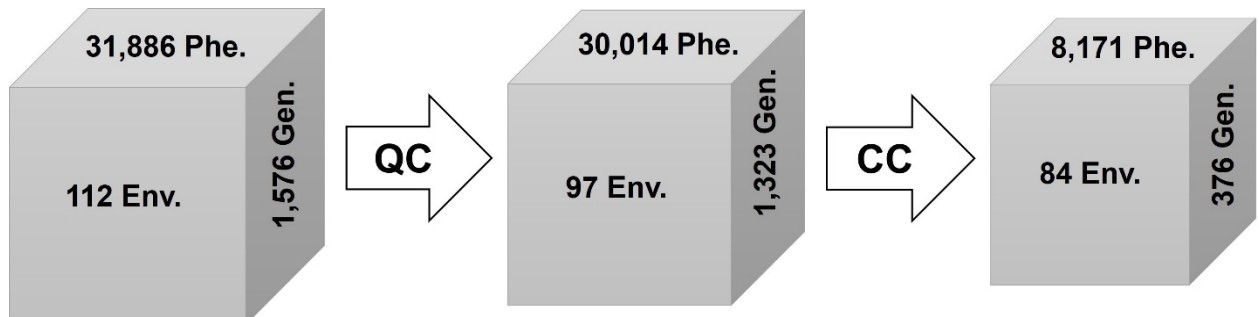

**Figure 9. The number of observations of G2F-Gen. (genomic data), G2F-Phe. (phenotypic data), and G2F-Env. (environmental data) in the original database, quality-controlled database, and the consistency-controlled database. The QC and CC refer to quality and consistency control algorithms.**

Following the FAIR principles, the multi-dimensional, consolidated, and enhanced G2F database along with developed

Python-based QC-CC scripts are released in the Zenodo platform for public access (findable and accessible). The associated documentation is also available for the database users. The folders and files structures are explained and interoperable, including the datasets preprocessing, the QC and CC sub-modules, and the implementation process for each G2F data release. Additionally, the database is usable for other crop growth modeling, and the scripts are modifiable to be implemented using datasets from other sources rather than G2F (reusable). The "CLIM4OMICS" database package along with the current study

can be taken as a guideline to create and enhance other databases with geospatial attributions for Earth System and crop growth modeling, and statistical analyses and learning.

The developed databases package in this study is a novel example of a multi-dimensional database with enhanced OMICs variables and improved hydroclimate data used in phenotype. Also, this novelty becomes relevant when the databases lacking genomic and phenotypic observations limit the use of multi-dimensional OMICs data for plant modeling (Germeier and Unger,
2019). Several databases, for example, Agricultural Model Intercomparison and Improvement Project (AgMIP), simulate agricultural risk under climate change, emphasizing environmental drivers, including weather and soil properties (AgMIP, 2022). The current developed database provides the interdisciplinary opportunity to integrate biological systems and climate science communities to benefit food security and resiliency applications in the changing anthropogenic climate.

Conessa and Beck (2019) and Persa et al. (2021) proposed and developed genomic and phenotypic quality control pipeline for
genomic selection that applies to any dataset. The current study's developed QC-CC framework for environmental drivers needs to include part of previous databases and algorithms that can play a tremendous role in crop phenotypes predictability. The enhanced G2F database version proposed here is known as "CLIM4OMICS," The data pre-processing framework is designed to interconnect the OMICs variables with climate drivers to improve the models' performance in the complex food systems.

**4.6. Error uncertainty**

In database creation and curation to successfully train and test crop growth models, the uncertainty quantification is a useful technique to assess the error sources. Quality and consistency controls enhance and consolidate multi-dimensional databases for achieving crop models high performance, and uncertainty assessment diagnose the main sources of error propagation in the models predictive skill.
The use of external databases (e.g., NSRDB, DayMet, and NWS) to impute and simulate missing environmental data propagates errors in sampling, modeling, and transforming environmental estimations into the G2F time series. These errors in the input data also propagate uncertainties into crop growth model outputs, which require the quantification of input data uncertainty. The standard uncertainty of the climate variables has been quantified using the SD of the PDF of the errors between the observed G2F time series and those of the external databases for a given climatic variable. For G2F improvement, the error
SD represents the uncertainty introduced by using each external data source (Steiner et al., 2013). Thus, first, we calculated the errors using Eq. (1), and then the PDFs of errors. The SD statistics of the error terms are then calculated (see Fig. 10).





(a)

(b)

(c)

(d)

(e)

(f)

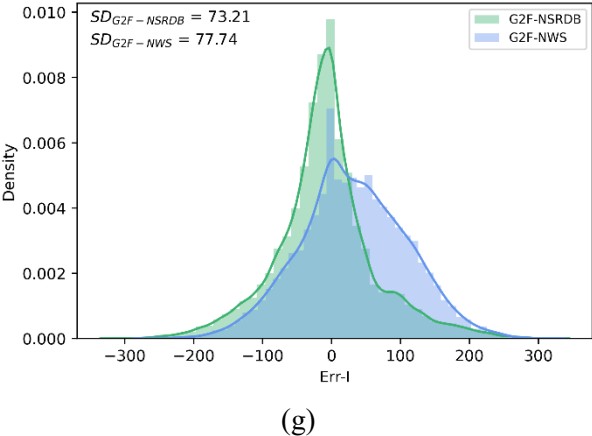

(g)

**Figure 10.** The probability distribution function (PDF) of the error values for (a) temperature (Err-T), (b) dew point (Err-D), (c) relative humidity (Err-H), (d) solar radiation (Err-S), (e) rainfall (Err-R), (f) wind speed (Err-W), and (g) wind direction (Err-I). Note that each of the external environmental data sources may not contain all the G2F hydroclimatic variables. The error term has been calculated for the common variables between G2F and each of the data sources. The SDG2F-NSRDB denotes the standard deviation of the errors between G2F and NSRDB, the SDG2F-DayMet denotes the standard deviation of the errors between G2F and DayMet, and SDG2F-NWS denotes the standard deviation of the errors between G2F and NWS for a given climatic variable.

Standard uncertainty is a very informative measurement when the PDF of errors is close to a normal distribution with a mean of zero (Merchant et al., 2017). Here, the error distribution for temperature (Fig. 10a), dew point (Fig. 10b), relative humidity
(Fig. 10c), rainfall (Fig. 10e), and wind direction (Fig. 10g) are normal. In the case of solar radiation, the normal distribution is reasonably fitted to the errors between G2F and NSRDB. Also, the PDF of the errors in wind speed are close to a normal distribution.

The SD has been calculated for the errors between G2F and each NSRDB, DayMet, and NWS databases. In the case of temperature, the smallest standard uncertainty of errors is obtained from DayMet (SD$_{\text{G2F-DayMet}}$ = 1.54). For dew point, the
NSRDB introduces the smallest error uncertainty (SD$_{\text{G2F-NSRDB}}$ = 2.39). In the case of relative humidity, although the SD statistics are very close for both NSRDB and NWS, it is slightly smaller for NWS (SD$_{\text{G2F-NWS}}$ = 11.17). For solar radiation, the uncertainty of using NSRDB to impute the gaps of G2F is considerably smaller than using DayMet (SD$_{\text{G2F-NSRDB}}$ = 51.37). The mean of errors for rainfall from both DayMet and NWS are close to zero. However, the error dispersion is substantially small for the DayMet (SD$_{\text{G2F-DayMet}}$ = 10.6). There is not a consistent pattern of uncertainty for the wind properties. For the
wind speed, the SD is slightly smaller from the NWS (SD$_{\text{G2F-NWS}}$ = 1.57), while in the case of the wind direction, NSRDB represents the smaller error uncertainty (SD$_{\text{G2F-NSRDB}}$ = 73.21). These SD statistics values illustrate the error magnitude introduced by using external databases. In case of using any other data sources rather than those provided by the G2F initiative, the uncertainty estimations show the sources of error propagation through the crop growth prediction.

By comparing all the error dispersion statistics for each climate variable, the minimum standard uncertainty is observed in
temperature (SD$_{\text{G2F-DayMet}}$ = 1.54), while the maximum uncertainty is observed in rainfall (SD$_{\text{G2F-NWS}}$ = 90.51). These results
are aligned with several previous studies that show rainfall as a complex phenomenon difficult to measure, model, and predict. This difficulty in rainfall estimates can also be attributed to the spatiotemporal heterogeneity of the collected data (Bruno et al., 2014; Pollock et al., 2018). However, the considerably small uncertainty of errors between G2F and DayMet for rainfall ($SD_{G2F-DayMet}$ = 10.6) illustrate the higher robustness of gridded databases (i.e., DayMet) and their usefulness to complement
in-situ databases (i.e., NWS) for improving the G2F-E datasets.

Note that the NWS is the only database that records wind gust. However, we removed the wind gust from the G2F-G database due to several missing values available in that database.

## 5. Data availability

The data that support the findings of this study "CLImate for Maize OMICS: CLIM4OMICS Analytics and Database" are
openly available in "Zenodo" at   http://doi.org/10.5281/zenodo.7490246. A quick guideline for performing the Python scripts is provided in "ReadMe.txt" file, and the required Python packages to be installed are listed in "Requirements.txt" file in the database package (Sarzaeim, et al., 2023).

## 6. Conclusion

In this study, we proposed an algorithmic QC-CC framework for data pre-processing pipeline to consolidate a homogeneous,
multi-dimensional, and enhanced database consisting of (1) OMICs observations, (2) hydroclimatic variables, and (3) metadata for statistical, data-driven, and biophysical crop growth models' applications to simulate GxE interaction. The G2F initiative database for maize phenotypes predictability across the U.S. and Ontario in Canada between 2014 and 2017 has been used to test the designed QC-CC framework. A QC sub-module has been developed for each G2F data dimension, including G2F-G, G2F-P, G2F-E, and G2F-M sub-modules. Each sub-module generally aims to (1) read the raw files, (2) check and correct
structural and format inconsistencies, (3) detect the missing values, and (4) fulfill them. The CC module is the last step of the input data pre-processing. It is designed to check the compatibility of controlled input data to identify the intersection of the records between all data dimensions ready for GxE model implementation and analytical operation. Multiple external data sources, including NSRDB, DayMet, and NWS, have been used to simulate the G2F-E gaps. The error uncertainty introduced by these data sources is also quantified.
After passing through the QC-CC data pre-processing pipeline, the structural inconsistencies have been corrected, and the missing values have been filled in G2F-G and G2F-E datasets. As a result, 84 G2F trials for GxE simulation are released, consisting of molecular genetic markers of 376 maize lines and 8,171 yield observations. Here, the target phenotypic observation is yield. However, other phenotypes like plant height, ear height, and grain moisture also have been provided in the improved database for users. The improved G2F-E database contains seven hydroclimatic time series during the maize
growing season in the G2F trial sites: temperature, dew point, relative humidity, solar radiation, rainfall, and wind speed and

direction. The proposed methodology is applicable for other spatiotemporal variables improvement for the GxE models implementation. The improved multi-dimensional G2F database, along with developed scripts in a Python environment, is freely available for all users to be employed in their research.

The database provided in this study can foster further efforts to improve GxE analytics and phenotypic predictability by
enhancing the quality and consistency controls robustness as listed below:

1. Employ remote sensing imageries to simulate and fulfill the crop's phenotypic missing values to involve more samples in the database and analytics of maize growth predictability,

2. Integrate other hydroclimate time series to provide a wide range of environmental drivers of maize growth for the improvement of GxE models' predictive skill, and

3. Develop rapid-response and user-friendly software architectures benefiting from pattern recognition techniques to correct typos, erroneous values, and data structure inconsistencies for boosting database management, analytical tools, and visualization efficiency.

## 7. Author contributions

PS and FMA designed and conceptualized the study idea and methodology. PS, HA, and FMA designed, processed, and
developed the datasets and scripts. PS and FMA prepared the original draft and reviewed it. DJ and NDLG contributed the development of the original G2F OMICs database and reviewed the draft.

## 8. Competing interests

The corresponding author has declared that none of the authors has any competing interests.

## 9. Acknowledgements

The authors acknowledge the support provided by the Agriculture and Food Research Initiative Grant number NEB-21-176 and NEB-21-166 from the USDA National Institute of Food and Agriculture, Plant Health and Production and Plant Products: Plant Breeding for Agricultural Production. Also, we thank the Genomes to Fields (G2F) Initiative for providing the experimental platform that created the original database. We are grateful to the UNL Holland Computer Center for access to their high-computing facilities to perform the analysis. We also acknowledge the support from Quantifying Life Sciences
Initiative at the University of Nebraska-Lincoln.



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
