# Peer review of "CLIM4OMICS: a geospatially comprehensive climate and multi-OMICS database for Maize phenotype predictability in the U.S. and Canada"

_Earth System Science Data, 2023_

## Author Response (AR1)

**Reviewer 1:**
**General comment:**
This study performed quality control (QC) and consistency control (CC) to identify and cure missing data in an existing dataset from Genomes to Fields (G2F) initiative. The manuscript provides detailed information on the procedures of applying the QC-CC to pre-process four sub-datasets: genomic (G), environmental (E), phenotypic (P), and metadata for each field trial (M). The program developed in this study can be useful to control input data quality for GxE model implementation. However, it is not clear to me that this is a significant advancement in the data, science, methods, or outcomes to warrant publication in this Journal.

**Reply:** We appreciate all your insightful comments. We have reviewed each comment and addressed them as suggested or responded to them as needed. In the revised manuscript we highlighted the novel contributions of coupling climate and OMICS data. Such coupling provides accessible digital resources for both climate and OMICS communities and will eventually trigger the advancement of climate and phenotype predictions (in addition to the progresses made on phenotype forecast). To the best of our knowledge the digital products, narratives, and selected literature in the reviewed manuscript present "…original research data (sets), furthering the reuse of high-quality data of benefit to Earth system sciences" (as stated in ESSD's aim).

On the other hand, you will notice that some of our responses will mention a timespan of eight years (2014-2021) instead of the original four years. The expansion of the dataset followed the suggestion of Reviewer 3. The narratives, Tables, and Figures were updated in the manuscript or mentioned in the responses when necessary. For your reference, we are attaching two files containing the updated and new Figures and Tables, including two figures to be added in supplementary materials. As a product of the expansion, we were able to include three experiments for the state of Niedersachsen in Germany (2018, 2020, and 2021).

*Major concerns:*
1.   The manuscript reads more like a manual instead of a scientific paper. There is plenty of information on how to apply QC step by step, however, there are few results of the developed data. Even in Section 4. Results and discussion, I can only find some examples of how the dataset looks in a table. Although it is geospatial data, there are no maps presenting the spatial pattern of data values. I cannot evaluate the data without the figures presenting the spatial and temporal changes in the data.
*Reply: Thanks for your comment. We address one of your points by highlighting the updates of the previous and new datasets through the manuscript published in ZENODO at http://doi.org/10.5281/zenodo.7490246 and http://doi.org/10.5281/zenodo.8060807, respectively. Also, we followed the Earth System Science Data aim that states: "Articles in the data section may pertain to the planning, instrumentation, and execution of experiments or collection of data, and any interpretation of data is outside of the scope of regular articles". We consider that the detailed explanation of the quality and consistency controls responds to ESSD's expectations and benefits data reusability. Also, we provided a thorough explanation of the quality and consistency control pipeline to address the issues present in the original G2F*

*data files (or databases of this kind). We anticipate that the developed pipeline could be a digital resource for other researchers to improve their own databases.*

*On your second point about the examples in the "Results and discussion" section, we included few examples of the improved data for each dimension as a showcase of how the data are presented in the released files in the database package in ZENODO.*

*Regarding the spatiotemporal visualization, we updated Figure 1 and added three Figures illustrating the geospatially distributed records. Figure 1 in the original text was updated by adding a map within the conceptual framework with the spatial distribution of G2F locations and the number of experiments per state in the US and Ontario in Canada. A new figure, Figure 4, was added containing the spatial distribution of number of phenotypic samples for the G2F experimental sites each year between 2014 and 2021. The second figure added is Figure 5 and it represents the spatial distribution of averaged temperature and precipitation in the area of study. The third figure added, Figure 6, provides a heatmap of the number of experiments and average of each hydroclimatic variable for the experimental fields by state or province between 2014 and 2021. Please, see the captions below:*

*Updated*

*Figure 1. A conceptual framework of quality and consistency control algorithms for the multidimensional Genomes to Fields (G2F) OMICs and hydroclimatic database. "G2F-G" denotes G2F genomic data, "G2F-P" denotes G2F phenotypic data, "G2F-M" denotes G2F metadata, and "G2F-E" denotes G2F environmental data. The map indicates the locations and number of sites per state used in by the G2F initiative and represented in the CLIM4OMICS (the map is expanded as Supplementary Figure 1).*

*Added*

*Figure 4. The spatial distribution of phenotypic records of G2F experiments in the U.S. regions and the province of Ontario in Canada between 2014. And 2021. The state of Niedersachsen in Germany includes the years 2018, 2020, and 2021 for three locations. The location of each station in the map was modified for visualization purposes, allowing the illustration of sations with multi-year records. The size of the circle represents the number of years sampled, which also appears within the parenthesis next to the year at each site. The colors of the circles were included for visualization purposes only.*

*Figure 5. The spatial distribution of (a) improved mean temperature (Tmean) and (b) improved accumulated rainfall (Racc) records in G2F-E database during the maize growing season in all G2F experimental fields in 2014-2017.*

*Figure 6. The heatmap for number of G2F experiments in the U.S. regions and the province of Ontario in Canada between 2014. And 2021. The state of Niedersachsen in Germany includes the years 2018, 2020, and 2021 for three locations. The color shows the number of stations in each state. The number in each cell represents the average of hydroclimatic variables in each state including mean of Temperature (T), mean of Dew Point (D), mean of Relative Humidity (R), mean of Solar Radiation (S), accumulative Rainfall (R), mean of Wind Speed (W), and mean of Wind Direction (I).*

*Please, consult the attached file "Sarzaeim et al ESSD Updated-FIGURES TABLES" with all the figures and tables in order of appearance in the manuscript, including two supplementary figures.*

2. The introduction mainly provides the background of QC and CC methods, but fewer contents of maize phenotype. What is the importance of developing maize data? What is the progress of this dataset compared with other datasets?

*Reply: We developed the present dataset to create a comprehensive database that contributes to fulfilling the absence of accessible, large-scale, multiyear, and multi-environment datasets that hinder scientists' ability to improve and test models for improving traits in crops using genotype-environment interactions (Lawrence-Dill et al., 2019 and other authors cited through the manuscript). For example, we mentioned in the Introduction that "To better understand the functionality of maize genetics across various environments, engineers, researchers, and economists can utilize the G2F database for simulating phenotypic outcomes through statistical models like GxE interactions". In our study, we introduce an enhanced version of the G2F multi-dimensional database, including genetic, phenotypic, environmental, and metadata dimensions across several locations in the U.S. and Canada. This database is ready to be used by researchers for maize phenotypic modeling, prediction, and improvement. Furthermore, we foresee this dataset as a "bridge" between the Earth system and OMICS scientists, and the basis for forthcoming showcases for agriculture's mitigation, resilience, and adaptation to climate change.*

3. Quality control is the key point in this study, which means the major effort is improving data quality by deleting or curing missing data in the original G2F dataset. No new dataset is developed in this study. The QC and CC used in this study are also commonly used methods, and I don't think there is any improvement in the method.

*Reply: Thanks for your comment. The goal of this study is to introduce a comprehensive quality and consistency controls that lead to the enhancement of the original G2F database. As outlined in section 2, some drawbacks of the original G2F database are the structural inconsistencies among files from different dimensions and years, typos, and missing values. In this study, we presented a user-friendly pipeline in Python to read multi-dimensional data from multiple files, correct data formats and inconsistencies and detect and fulfill data gaps. The delivered digital products represent an opportunity for the climate science community to access extensive maize genetic data for the development and implementation of Earth System models in diagnostic, forecasting, and predictive frameworks. It also enables the genetic science community to incorporate various hydroclimatic data into maize phenotype predictive models. We believe the enhanced data package along with the developed Python scripts can benefit researchers to: (1) improve maize yield predictability using the data package; (2) adapt the script pipeline for quality and consistency controls for their own data sets; (3) promote data sharing and discoverability based on FAIR data principles; and (4) connect the climate and genotyping data scientists and communities.*

4. Using screenshots as figures (Figs 3-6) is not a good way to introduce data in a scientific paper. I suggest listing a table to only explain the head names in each sub-dataset.

*Reply: We appreciate your comment. We addressed your comment as follows:*
*We created tables instead of figures. Former Figures 3-6 are now Tables 1-4, respectively. While each table shows the header names for each sub-dataset in ZENODO, we kept some of the values for each heading as examples. This data illustration will allow the reader and data-user a more effective connection between the manuscript and the database in ZENODO. The captions contain information about the source file directories for the genomic, phenomic, environmental, and*

*meta data examples at "File Upload/Genotype/Markers.txt",*
*"FileUpload/Phenotype/g2f_2014_hybrid_data_clean.csv",*
*"FileUpload/Environment/g2f_2014_weather.csv", and*
*"FileUpload/Meta/g2f_2014_field_characteristics.csv" in the data package, respectively.*

[revised manuscript text omitted]

5.  "Maize phenotype predictability" is mentioned in the title, but I cannot find any work related to predictability. Building relationships between maize phenotypes and other environmental and genomic factors may improve this study.

**Reply:** *Thanks for your comment. This very relevant comment led us to re-assess the title, premises, and thesis used in the manuscript. We respond that the title suggests the application but does not declare an assessment. For example, a title involving the term "for improving GxE model predictive skill" requires us to create a new thesis about the expansion of climate, and OMICS data will lead to an improvement in the predictability of phenotypes. Such a thesis was*

*presented in Sarzaeim et al. (2022a), which led us to create this database. This point has been stated in the "Introduction" and "Results and discussion" sections, and it is key in the formulation and support of the thesis that comprehensive quality and consistency controls are useful for the development and consolidation of such an enhanced, high-quality, large-scale, and multi-dimensional database for maize phenotypes prediction. Further, while our current study focuses on the steps to enhance the input for maize phenotypic models, we added a brief explanation in maize yield predictability improvement using the enhanced version of database to the results section in the revised manuscript.*

*As a side note in this response but referenced in the manuscript: In our previous publication (Sarzaeim et al., 2022a, referenced in the manuscript), we have evidenced the application of these quality and consistency controls for high-dimensional databases to improve the statistical models' performance, such as the GxE by 12.1% in maize phenotypic predictability. In another to-be-submitted paper, we analyzed the sensitivity of maize phenotypes' predictability to hydroclimatic variables.*

**Minor:**

1. Section 2. How different dimensions of data connect. Which head is the key field in the database

**Reply:** *Thanks for the comment. As part of the "Consistency Control" module, the initial Environmental and Meta files IDs were referred to as "Experiments," containing State Abbreviation and Hybrid ID. Phenotype files IDs were called "Records." Section 4.5 describes the consistency control module's unique new ID for all dimensions. This unique ID is the key field conformed by the Year, State Abbreviation, and hybrid IDs and is shared among G2F-P, G2F-E, and G2F-M files. Additionally, G2F-P and G2F-G IDs share the "P1" and "P2" columns (or Phenotype Pedigree columns), all forming the G2F-G's GIDs.*

2. Line 211. Figs.

**Reply:** *Thanks for your comment. We addressed it in the revised manuscript.*

3. Caption in Figure 4. Repeated "shows".

**Reply:** *We removed the repeated word of "show" in the revised manuscript.*

4. Uncertainty. What about the errors in other sub-datasets, aside from climate?

*Reply: Thanks for your comment. In this study, we employ external hydroclimatic databases (explained in section 2.3.1.) to simulate the G2F environmental data gaps (explained in section 3.1.3.), which propagates errors in G2F environmental timeseries. Based on this method, the uncertainty quantification of the environmental data is necessary. For the other data dimensions, including G2F genetic and phenotypic data, we did not use any external databases for gap fulfillment in this research. Although uncertainty quantifications for genetic and phenotypic data are highly important, they are out of the scope of the current study.*

5. Table 1. What is the meaning of the "DEH1" under the "location" field? Do you think it is a good way to provide location information?

*Reply: Thanks for this question. Section 2.2. indicates the meaning of this 4-character name or"Field-Location" extracted from the G2F experiment. This alphanumeric value consists of the state's abbreviation in the two first characters and the name of the hybrid experiment in the last two characters tested in that state. Thus, "DEH1" shows that the experiment with the name of "H1" is sampled in Delaware. This naming method is the standard one that has been taken to indicate the location of the experiment (including the state and the experimental field) by the G2F database developers in original released files. Additionally, to avoid confusion, we have brought an example of naming in section 4.2. As the project and experimental sites expand, the addition of latitude and longitude values may substitute or complement the state's ID.*
* * *
**Reviewer 2:**
**General Comment:**
This manuscript developed a pipeline for processing a variety of data generated by Genomes to Fields (G2F) initiatives, and provided a comprehensive database for maize phenotype prediction. This study bridges the gap between data analytics and scientific research, meanwhile, advocates using G2F data for understanding maize biology. Although it's valuable, still some aspects can be improved.

*Reply: We appreciate your insightful comments on the manuscript. We have gone through each comment thoroughly and addressing them as suggested or responding to them as needed in the revised manuscript.*

*On the other hand, you will notice that some of our responses will mention a timespan of eight years (2014-2021) instead of the original four years. The expansion of the dataset followed the suggestion of Reviewer 3. The narratives, Tables, and Figures were updated in the manuscript or mentioned in the responses when necessary. For your reference, we are attaching two files containing the updated and new Figures and Tables, including two figures to be added in supplementary materials. As a product of the expansion, we were able to include three experiments for the state of Niedersachsen in Germany (2018, 2020, and 2021).*

*For your reference, you can consult the attached file "Sarzaeim et al ESSD Updated-FIGURES TABLES" with all the figures and tables in order of appearance in the manuscript, including two supplementary figures.*

(1) the genotype imputation approach recommended by G2F (line 242-244) is not tested and compared with other classical approaches, such as BEAGLE, IMPUTE.

*Reply: Thank you for your comment. We considered that the implemented naive imputation for each molecular marker was sufficient and an eventual trackable source of error in missing values imputed by the mean. This response acknowledges your comment and inspires the development of an approach analogous to the one developed by Sarzaeim et al. (2022) for genomic and phenomic data. In the new version of the manuscript, we re-wrote the lines above for clarity.*

(2) it might not be reasonable to keep only the intersection among data dimensions (line 601-602), because this data manipulation could cause information loss as numerous records are filtered out (Figure 9). It's proved that some records with genotypic data but no phenotypic or environmental data are also useful in genetic analysis.

*Reply: Thank you for your comment. We agree that genetic data, with the lack of phenotypic and environmental data, are beneficial for further research in genetic studies and even phenotypic modeling purposes. The purpose of CLIM4OMICS is to introduce a quality and consistency control criterion to be used and adapted by other researchers for GxE modeling and other statistical and biophysical modeling applications and analytics for maize phenotypes prediction. In such models, complete input data intersected among all the data dimensions and with no gaps is required. To this end, we proposed the consistency control module. We make accessible the resultant database, codes, and practices following FAIR principles. Further, we kept the original files containing all the data in the database package along with available Python scripts to be modified by the researchers to change the criteria for including desired samples or any other applications and analysis that does not require the intersection.*

(3) it's better to provide a case of applying G2F data processed by the pipeline to understand genotype by environment interaction in maize (line 16-18). It's even better if the scientific interpretation is easier for processed data than for unprocessed data.

*Reply: This study aims to consolidate an improved version of a multi-dimensional database, including genetic, phenotypic, environmental, and metadata files, to be ready for use by other researchers for further research in maize phenotype prediction and model development. While in the current study, we focus on high-quality data for modeling purposes, we have shown in our previous work (Sarzaeim et al., 2022a, referenced in the manuscript) how the improvement in data quality leads to improving the GxE models performance by 12.1% in maize yield predictability. This result is explained in the "Results" section in the revised manuscript.*

(4) The URL provided by the authors is not accessible (line 30-31).

*Reply: We checked the provided link and it works.*

(5) Line 376-382 are written repeatedly.

**Reply:** *We revised and removed it in the updated manuscript.*

(6) The quality of figures can be improved, such as Figure 2-6, and 10.

**Reply:** *As per suggestion of Reviewer 1, Figures 2-6 were transformed into Tables 1-4, respectively. Also, the quality of Figure 10 now Figure 7, has been improved in the revised manuscript. Also, per suggestion of Reviewer 1, two additional Figures were added (Figures 3-5) and Figure 1 updated, which captions are listed below and the figures are in the updated manuscript for your reference).*

*Updated*

*Figure 1. A conceptual framework of quality and consistency control algorithms for the multidimensional Genomes to Fields (G2F) OMICs and hydroclimatic database. "G2F-G" denotes G2F genomic data, "G2F-P" denotes G2F phenotypic data, "G2F-M" denotes G2F metadata, and "G2F-E" denotes G2F environmental data. The map indicates the locations and number of sites per state used in by the G2F initiative and represented in the CLIM4OMICS (the map is expanded as Supplementary Figure 1).*

*Added*

*Figure 4. The spatial distribution of phenotypic records of G2F experiments in the U.S. regions and the province of Ontario in Canada between 2014. And 2021. The state of Niedersachsen in Germany includes the years 2018, 2020, and 2021 for three locations. The location of each station in the map was modified for visualization purposes, allowing the illustration of sations with multi-year records. The size of the circle represents the number of years sampled, which also appears within the parenthesis next to the year at each site. The colors of the circles were included for visualization purposes only.*

*Figure 5. The spatial distribution of (a) improved mean temperature (Tmean) and (b) improved accumulated rainfall (Racc) records in G2F-E database during the maize growing season in all G2F experimental fields in 2014-2017.*

*Figure 6. The heatmap for number of G2F experiments in the U.S. regions and the province of Ontario in Canada between 2014. And 2021. The state of Niedersachsen in Germany includes the years 2018, 2020, and 2021 for three locations. The color shows the number of stations in each state. The number in each cell represents the average of hydroclimatic variables in each state including mean of Temperature (T), mean of Dew Point (D), mean of Relative Humidity (R), mean of Solar Radiation (S), accumulative Rainfall (R), mean of Wind Speed (W), and mean of Wind Direction (I).*

*Transformed Figures into Tables*

*Table 1. Overview of raw G2F-G data illustrating the genotyping by sequencing the molecular marker sequences of different hybrids stored in a single HDF-format file. The first column shows the maize hybrid codes, and the first row shows the locus information. The A, T, G, C, and R letters are a sample of the major and minor alleles at different marker positions. The letter N denotes the missing markers in a genetic sequence at each molecular site. The source file directory for the genetic data is in "File Upload/Genotype/Markers.txt" in the database package.*

*Table 2. Overview of the raw G2F-P data stored in ".csv" file format showing detailed information of the phenotypic observations in 2014 as one example of the multi-year data. The "Year" column shows the year of the a specific G2F experiment, "Field-Location" column shows the 4-character name of G2F experiment consisting of the state abbreviation in the two first characters and the name of the hybrid experiment in the last two characters tested in that state, the "Recid" column shows the ID of the phenotypic record, the "Source" column shows the source of the collected phenotypic sample portal, the "Plant Height [cm]" column shows the height of the plant in [cm], the "Ear height [cm]" column shows the height of the ear in [cm], the*

"Stand Count [plants]" column shows the number of plants per plot at harvest, the "Root Lodging [plants]" column shows the number of plants that show the root lodging per plot, the "Stalk Lodging [plants]"column shows the number of broken plants per plot at harvest, and the "Grain Moisture [%]" column shows the percentage of the water content in plant at harvest. The other phenotypic variables have been measured and stored in similar columns. The blank cells represent the missing values of phenotypic observations. The source file directory for the phenotypic data example is in "File Upload/Phenotype/g2f_2014_hybrid_data_clean.csv" in the database package.

Table 3. Overview of raw G2F-E data stored in ".csv" file format showing the environmental time series in tabular dormat for 2014 as one example of the multi-year data. The "Record Number" column shows the number of weather station records in each experiment, the "Experiment" column shows the 4-character name of G2F experiment consisting of the state abbreviation in the two first characters and the name of the hybrid experiment in the last two characters tested in that state, the "Station ID" column shows the ID of the weather station, "NWS Network" and "NWS Station" columns show the nearest NWS network and station has been used for initial QC by the G2F collaborators, the "Day [Local]", "Month [Local]", "Year [Local]", and "Day of Year [Local]" columns show the local day, month, year, and day of year of the weather record, "Daytime [UTC]" column shows the coordinated universal time, "Temperature [C]", "Dew Point [C]", "Relative Humidity [%]", "Solar Radiation [W m$^2$]", " Rainfall [mm]", " Wind Speed [m s$^{-1}$]", Wind Direction [degrees]", and "Wind Gust [m s$^{-1}$] column shows the hydroclimatic time series. The blank cells represent the missing values of phenotypic observations. The source file directory for the environmental data example is in "File Upload/Environment/g2f_2014_weather.csv", in the database package.

Table 4. Overview of raw G2F-M data stored in ".csv" file format showing the metadata collected for the 2014 experiments as one example of the multi-year data. The "Location Name" column shows the state and the number of the experiment in that state, the "Type" column shows the type of the experiment which can be hybrid or inbred, the "Experiment" column shows the 4-character name of G2F experiment consisting of the state abbreviation in the two first characters and the name of the hybrid experiment in the last two characters tested in that state, the "City" column shows the city that the experiment located at, the "Farm" column shows the name of the farm that the experiment has been tested in, the "Field" column shows the name of the field of the experiment, and "lon" and "lat" columns show the longitude and the latitude of the weather station installed in the field. The source file directory for the metadata example is in "File Upload/Meta/g2f_2014_field_characteristics.csv" in the database package.
* * *
**Reviewer 3:**
**General Comment:**

This study is expected to be useful to those studying maize (especially those interested in GxE effects) and provides a useful workflow for cleaning large, multi-factor datasets, for those working in other species. This is promoted by making the data processing scripts public and including details on execution order and library versions to be used. Transparency and repeatability in such a processing pipeline is key to allow for others to modify the workflow to accommodate additional data or otherwise customize it to suit their needs. Furthermore, access to common datasets is valuable for the purpose of benchmarking new methods and training students. However, there appear to be limitations which may hinder adoption and reuse, which seems to be a key aspect of this work.

***Reply:*** *We appreciate your insightful comments on the manuscript. We have gone through each comment thoroughly and addressing them as suggested or responding to them as needed in the revised manuscript. You will notice that some of our responses below will include a timespan of eight years (2014-2021) as pre your suggestion. The narratives, Tables, and Figures were updated in the manuscript or mentioned in the responses when necessary. For your reference, we are attaching two files containing the updated and new Figures and Tables, including two figures to be added in supplementary materials. As a product of the expansion, we were able to include three experiments for the state of Niedersachsen in Germany (2018, 2020, and 2021). For your reference, you can consult the attached file "Sarzaeim et al ESSD Updated-FIGURES TABLES" with all the figures and tables in order of appearance in the manuscript, including two*

*supplementary figures. Below is a list of updated and added Figures, and transformed Figures into Tables as per suggestion of one of the reviewers.*

*Updated*

*Figure 1. A conceptual framework of quality and consistency control algorithms for the multidimensional Genomes to Fields (G2F) OMICs and hydroclimatic database. "G2F-G" denotes G2F genomic data, "G2F-P" denotes G2F phenotypic data, "G2F-M" denotes G2F metadata, and "G2F-E" denotes G2F environmental data. The map indicates the locations and number of sites per state used in by the G2F initiative and represented in the CLIM4OMICS (the map is expanded as Supplementary Figure 1).*

*Added*

*Figure 4. The spatial distribution of phenotypic records of G2F experiments in the U.S. regions and the province of Ontario in Canada between 2014. And 2021. The state of Niedersachsen in Germany includes the years 2018, 2020, and 2021 for three locations. The location of each station in the map was modified for visualization purposes, allowing the illustration of sations with multi-year records. The size of the circle represents the number of years sampled, which also appears within the parenthesis next to the year at each site. The colors of the circles were included for visualization purposes only.*

*Figure 5. The spatial distribution of (a) improved mean temperature (Tmean) and (b) improved accumulated rainfall (Racc) records in G2F-E database during the maize growing season in all G2F experimental fields in 2014-2017.*

*Figure 6. The heatmap for number of G2F experiments in the U.S. regions and the province of Ontario in Canada between 2014. And 2021. The state of Niedersachsen in Germany includes the years 2018, 2020, and 2021 for three locations. The color shows the number of stations in each state. The number in each cell represents the average of hydroclimatic variables in each state including mean of Temperature (T), mean of Dew Point (D), mean of Relative Humidity (R), mean of Solar Radiation (S), accumulative Rainfall (R), mean of Wind Speed (W), and mean of Wind Direction (I).*

**Transformed Figures into Tables**

[revised manuscript text omitted]

**Major Concerns Regarding:**

**1. Data coverage relative to release date**

It appears that at present only data from 2014-2017 are included. As of writing, the Genomes to Fields Initiative has publicly released data up to 2021 so perhaps half the available observations are not included
https://datacommons.cyverse.org/browse/iplant/home/shared/commons_repo/curated/GenomesToFields_G2F_data_2021. I recommend inclusion of these data which would serve two functions.:

1. It would increase the likelihood of others using this resource by vastly increasing the available information for use. Not only in terms of phenotypic observations and locations but also in terms of additional genotypic data (the 2018 release contains additional entries https://datacommons.cyverse.org/browse/iplant/home/shared/commons_repo/curated/GenomesToFields_G2F_Data_2018/d._2018_genotypic_data )
2. It would serve as a test case for the durable utility of this workflow. By incorporating additional data, the authors would be able to comment on the expected time and resources required for incorporating additional data.

Alternatively, the authors could discuss motivating reasons for excluding the 2018-2021 data.

**Reply:** *Thank you for your valuable comment. We updated the entire dataset and the manuscript to cover all available data up to 2021. the first version (v1.0) updated data (or v2.0) is accessible in ZENODO at* [http://doi.org/10.5281/zenodo.7490246](http://doi.org/10.5281/zenodo.7490246) *and* [http://doi.org/10.5281/zenodo.8060807](http://doi.org/10.5281/zenodo.8060807), *respectively.*

**2. Demonstration of utility**

Quantification of the benefits of these methods would be a nice addition to highlight the value this work provides. This could be as simple as a table showing the number of usable observations or measurements with and without standardization/imputation, potentially with the differences in statistical power indicated. A more complicated example could be plotting several results with outlier filtering or imputation.

*Reply: Thanks for your comment. In the former Figure 9 (Figure 7 in the revised manuscript), we demonstrated the number of observations usable for simulating the GxE interactions after applying the QC-CC pipeline. In our previous work (Sarzaeim et al., 2022a, referenced in the manuscript) we showed that the enhancement of the G2F data increases the number of useable and complete experiments for GxE modeling. The increase of the number of experiments, from 32 to 84, using data of 2014-2017 benefited the maize yield predictability. Statistics of efficiency like the coefficient of determination ($R^2$) increased by 12.1%, the Root Mean Squared Error (RMSE) by 2.2%, the Mean Squared Error (MSE) by 11.4%, and the Mean Absolute Error (MAE) by 1.4%. This description was added to the revised manuscript to show the benefits of QC-CC pipeline in maize yield predictability in the revised manuscript.*

**3. Specific Text:**

- Line 297-8 "Also, the user needs to fix the typos manually in the raw files, otherwise the file is ready for the next control step." Are the updated names tracked or reported? It seems that by manually altering the original data (instead of treating it as immutable and saving the original and corrected values in a separate file) reproducibility relies on the operator making the same changes to the original data. This could be addressed by including a text file of the changes made manually.

  *Reply: The inconsistencies in the original G2F database are different. It means that the typos are not the same in different files and years, and consequently the required changes in the files are not the same. The Python script detects the typos in each file and provides the details of what exactly needs to be changed and how. Thus, each available typo issue is considered a unique one that needs to be corrected. The creation of a log file has been added from the typos' detection script. This will keep the record of the eventually modified data. While it was not contemplated for this study, we will explore the use of pattern recognition techniques to automatize the characterization and correction of typos. However, this effort will be included in future updates or manuscripts.*

- Line 565 "(Fig. 10c), rainfall (Fig. 10e), and wind direction (Fig. 10g) are normal." Was a test used to assess normality here? If not, please alter to "roughly normal", "approximately normal" or similar.

  *Reply: Thank you. We replaced the "approximately normal" to "roughly normal" in the revised manuscript.*

**Minor Concerns Regarding:**

**1. The Code:**

- I would recommend releasing a version of this package on GitHub or a similar service. This would support other researchers forking the repository to customize the workflow

(or adapt it to a different crop), issuing pull requests to add features or processing for related datasets and future data releases.

**Reply:** *Thanks for your comment. The script package has been released on GitHub at this link*: https://github.com/HasnatJutt/CLImate-for-Maize-OMICS_CLIM4OMICS-Analytics-and-Database

- There was at least one API key included. You may want to remove those and/or deactivate them.

  **Reply:** *We have deactivated the API keys.*

- Are there supposed to be duplicate files in Database/code? E.g. S_Database and S_Database (1) ?

  **Reply:** *Thanks for your comment. We have corrected the inconsistency in the updated data package.*

2. **The Figures:**
   - Figure 2. I would recommend using a text excerpt rather than screenshot to avoid pixilation.

     **Reply:** *We replaced the screenshot with the text in the revised manuscript.*

   - Figure 3. In the

     **Reply:** *Thanks for your comment. We addressed the editing comments in the entire manuscript.*

   - Figures 4-6. These screenshots have inconsistent numbers of rows included and (in the case of figure 4) have some columns collapsed (columns A-D and Z-AE are shown). I would recommend including the desired example rows in a table.

     **Reply:** *We totally agree that the number of rows are different. It is because of the differences in the format and data structure of the collected data. Each row in Figure 4 (Table 2 in the revised manuscript) shows a phenotypic single sample in a specific experiment for an individual maize line, while the recorded data in Figure 5 (Table 3*

*in the revised manuscript) is time series during the maize growing season. Figure 6 (Table 4 in the revised manuscript) also contains the metadata for a specific experiment-year in each row. In addition, the number of row and columns are too large to be able to be represented in a word document, thus we just illustrated a chunk of each data dimension. However, based on your recommendation, we converted the screenshots into table format for better quality of the data representation and displayed 15 rows of phenotypic and environmental datasets for consistency (Table 2 and 3 in the revised manuscript). These tables are exactly from the source files stored in "File Upload" folder in the database package.*